# Functional heterogeneity of lymphocytic patterns in primary melanoma dissected through single-cell multiplexing

**Francesca Maria Bosisio[1,2]†*, Asier Antoranz[3,4]†*, Yannick van Herck[5], Maddalena Maria Bolognesi[2], Lukas Marcelis[1], Clizia Chinello[2], Jasper Wouters[1], Fulvio Magni[2], Leonidas Alexopoulos[3,4], Marguerite Stas[6], Veerle Boecxstaens[6], Oliver Bechter[5], Giorgio Cattoretti[2], Joost van den Oord[1]**

[1]Laboratory of Translational Cell and Tissue Research, KU Leuven, Leuven, Belgium; [2]Pathology, Department of Medicine & Surgery, Università degli studi di Milano-Bicocca, Milan, Italy; [3]ProtATonce Ltd, Athens, Greece; [4]National Technical University of Athens, Athens, Greece; [5]Laboratory of Experimental Oncology, KU Leuven, Leuven, Belgium; [6]Department of Surgical Oncology, KU Leuven, Leuven, Belgium

**Abstract** In melanoma, the lymphocytic infiltrate is a prognostic parameter classified morphologically into 'brisk', 'non-brisk' and 'absent' entailing a functional association that has never been proved. Recently, it has been shown that lymphocytic populations can be very heterogeneous, and that anti-PD-1 immunotherapy supports activated T cells. Here, we characterize the immune landscape in primary melanoma by high-dimensional single-cell multiplex analysis in tissue sections (MILAN technique) followed by image analysis, RT-PCR and shotgun proteomics. We observed that the brisk and non-brisk patterns are heterogeneous functional categories that can be further sub-classified into active, transitional or exhausted. The classification of primary melanomas based on the functional paradigm also shows correlation with spontaneous regression, and an improved prognostic value when compared to that of the brisk classification. Finally, the main inflammatory cell subpopulations that are present in the microenvironment associated with activation and exhaustion and their spatial relationships are described using neighbourhood analysis.

***For correspondence:**
f.bosisio1@gmail.com (FMB);
asierantoranz91@gmail.com (AA)

†These authors contributed equally to this work

## Introduction

The lymphocytic infiltrate in melanoma is a prognostic parameter reported by the pathologist as patterns of tumour-infiltrating lymphocytes (TILs). The 'brisk' pattern (diffuse or complete peripheral TILs infiltration) has a better prognosis compared to the 'non-brisk' (tumour areas with TILs alternate with areas without TILs) or to the 'absent' pattern (no TILs or no contact with melanoma cells) (*Clark et al., 1989*; *Clemente et al., 1996*; *Mihm et al., 1996*). There are multiple pitfalls in the purely morphological evaluation of TILs (*Bosisio and van den Oord, 2017*), but the most important one is that morphology alone cannot determine their activation status. The meaning of the word 'brisk' according to the dictionary is 'active, energetic', a definition implying a functional connotation starting from a morphological evaluation. Surprisingly, this functional connotation has never been proved. The contact between cytotoxic lymphocytes (Tcy) and melanoma does not always lead to tumor eradication but, due to immune modulation, can also result in Tcy inactivation. These 'exhausted' Tcy would still be morphologically present, indistinguishable from active lymphocytes. Moreover, the morphological (*Saltz et al., 2018*) and functional (*Krieg et al., 2018*) side of the

tumour microenvironment has been separately investigated, but integration of both types of data is still lacking.

In recent years, several methods for single-cell analysis have been implemented in order to obtain a high-resolution landscape of the tumour microenvironment (*Anon, 2017*). According to a recent review (*Binnewies et al., 2018*), high resolution means to characterize not only the immune infiltrate but also to define the spatial distribution of each component within it, which allows one to make inferences about cell-cell interactions. Nevertheless, most of these methods rely on dissociation of the cells from fresh material, an impractical option in primary melanomas, nowadays diagnosed at an early stage, with very limited material, and resulting in loss of the knowledge of the spatial distribution of each component within the tumor. Moreover, several studies for predictive biomarkers relies on peripheral blood (*Ascierto et al., 2017*). Though, the functional status of circulating inflammatory cells can completely change entering the tumour site (*Buggert et al., 2018*). Intuitively, it is the behaviour of the inflammatory cells in the surroundings of the tumour that will make a difference in terms of prognosis and response to therapy.

Here, we characterize the immune landscape at single cell-level in primary melanoma based on a panel of 39 immune markers applied on one single tissue section through a high-dimensional multiplexing method (*Cattoretti et al., 2001*; *Bolognesi et al., 2017*), a RT-PCR expression evaluation and a shotgun proteomic analysis. This approach allowed us (a) to further categorize the brisk and non-brisk morphological patterns of TILs into three functional categories; (b) to define the correlation between T-cell activation and spontaneous melanoma regression; (c) to investigate the most important inflammatory subpopulations involved in TILs exhaustion (*Figure 1*).

## Results

### Functional analysis of TILs

We classified each CD8+ cell in the TMA cores as part of a spectrum ('functional status') ranging from 'active' (CD69high and/or OX40high) to 'exhausted' (TIM3highCD69lowOX40low) (*Figure 2A–D* and Supplementary Data 1). We observed that the rotation vector of LAG3 was not aligned to TIM3. Moreover, very few cells expressed LAG3, therefore this marker had a very small impact on exhaustion. The assignation of a functional status to each core in the TMA ('core status', see Materials and methods) yielded 17/60 cores defined as active, 23/60 in transition, and 20/60 defined as exhausted. Core classification allowed the assessment of the heterogeneity of the immune response in different areas of the melanoma for the same patient. From the 29 patients included in the analysis, eight patients allowed to sample only a single core due to the size of the melanoma and could not be included in this analysis. From the 21 remaining patients, 10/21 showed homogeneous core statuses: four active, five in transition, and one exhausted; and 11/21 showed heterogeneity. Correlation with clinical survival (overall survival, OS) showed that patient classification based on functional status has an improved prognostic performance (log-rank p.value = 0.079) when compared with the brisk morphological classification (log-rank p.value = 0.36) (*Figure 2E*). Also repeating the survival analysis in the SKCM TCGA data set confirmed that the patients in the 'Active' group had better prognosis than the patients in the 'Exhausted' group (log-rank p-value=0.0082) validating the results obtained with our dataset (*Figure 2—figure supplement 1*).

We then checked whether the core status was significantly associated with spontaneous regression of the tumour, regarded as the result of a successful Tcy immune response, and with other histopathological parameters (histological subtype, ulceration, Breslow thickness, mitoses). Late regression areas indeed showed significant differences in the mean level of activation of the cores as compared to early regression (p=0.022) and no regression (p=0.031). No significant differences were instead found between early regression and no regression (*Figure 3A,B*). Higher levels of activation were found in Lentigo Maligna Melanoma (p=0.02). However, since only three LMM cores from two patients were included in our data set, no definite conclusions can be drawn from this data. The other histopathological prognostic parameters did not show significance (*Table 1*).

### Phenotypic identification

The inflammatory subpopulations were identified using three different unsupervised clustering methods (KMeans, PhenoGraph, and ClusterX *Chen et al., 2016*) followed by manual annotation of the

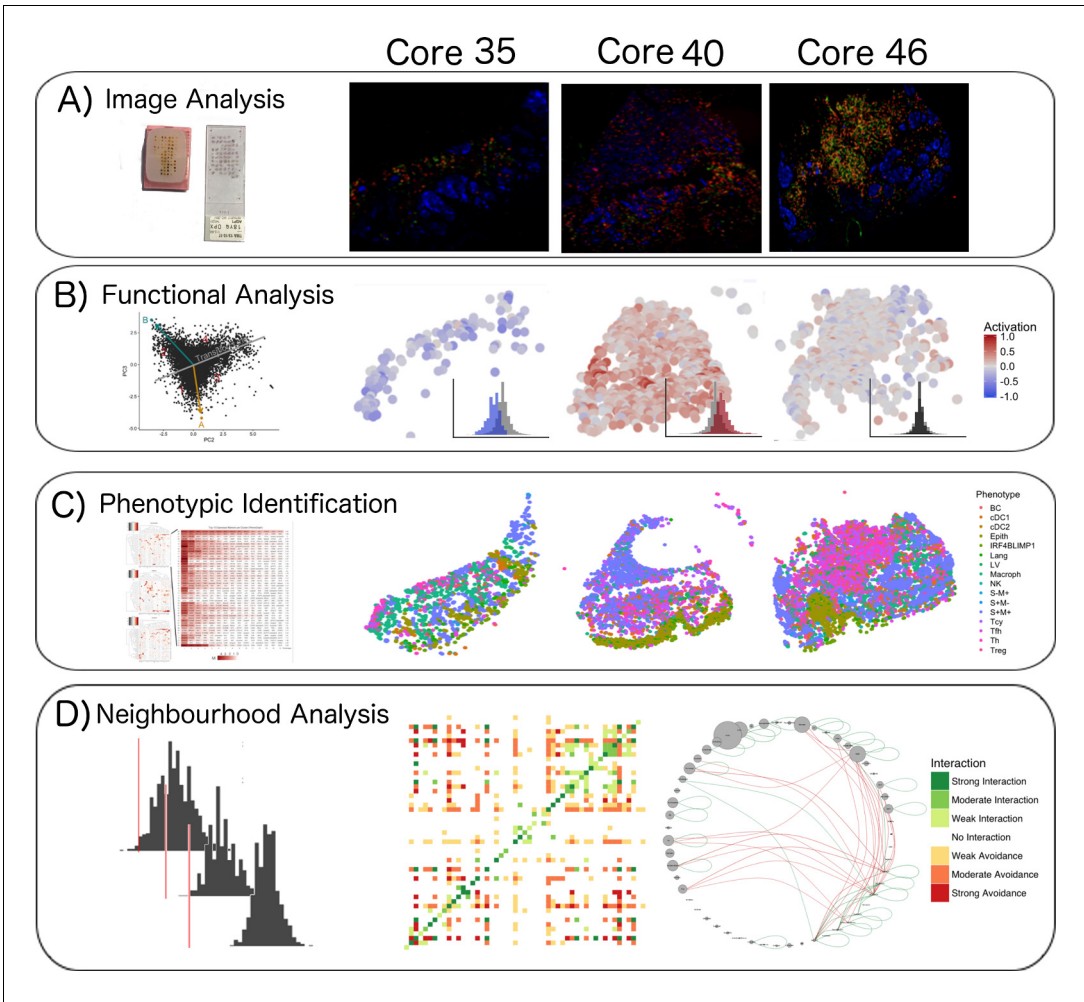

**Figure 1.** Single-cell analysis scheme. (**A**) TMA construction, Multiplex-stripping immunofluorescence: 60 cores were obtained from 29 patients, assembled in a Tissue Micro Array, and analysed using the MILAN technique; the immunofluorescence images from one round of staining (three markers/round: S100-blue, CD3-red, CD4-green) of three representative cores from our data set are shown. (**B**) CD8+ cells were analyzed using image analysis for functionality using an activation parameter derived from multiple activation and exhaustion markers evaluated at single-cell resolution; the CD8+ cells are here digitally reconstructed for each of the above standing cores, preserving their spatial distribution in the tissue section and assigning each of them a color according to the activation status. (**C**) All the cell populations in the cores were phenotypically identified using consensus between three clustering methods and manual annotation from the pathologists; The heatmaps with the levels of expression of the phenotypic markers per cluster for one of the three clustering methods are shown on the left; on the right, all the inflammatory cells are assigned a color based on their phenotype and the tissue is digitally reconstructed for each of the above standing cores, preserving the spatial distribution of each cell. (**D**) The social network of the cells was analysed using a permutation test for neighborhood analysis in order to make inferences on cell-cell interactions. The results of the neighborhood analysis are generated as a heatmap were the type and the strength of the interaction is expressed with a color code; to simplify the visualization of the interactions, the different cell types are represented in a circle and connected with lines that clarify the type of relationship between them.

clusters by an expert pathologist (FMB). The choice of the markers to identify the inflammatory cell populations is based on our previous review focusing on the melanoma microenvironment (*Bosisio and van den Oord, 2017*). Cells were evaluated for consistent cell phenotype as described in the Materials and methods (*Figure 4—figure supplement 1*). From the 19 clusters identified, 17 could be associated to specific cell lineages, while the remaining two were discarded. Based on the inclusion criteria described in the methods, 179,304 out of 242,224 cells (74.02%) were included for

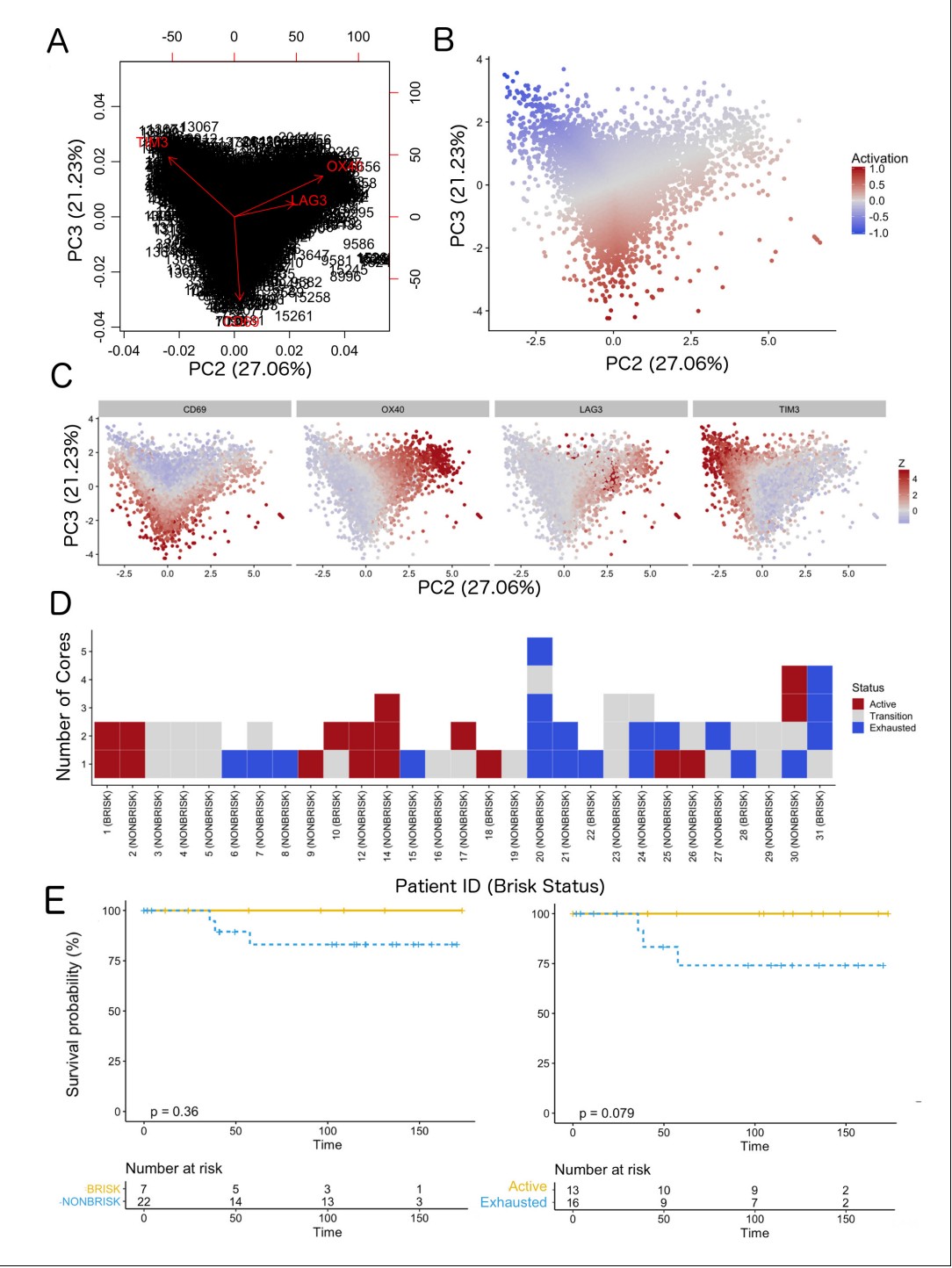

**Figure 2.** Definition of activation and implications in overall survival. A biplot showing the projection of the cells as well as the rotation vectors of the markers over PC2 and PC3 has been created using only CD8-positive cells and the four markers relevant for their functional status: CD69, OX40, LAG3, and TIM3. (**A**). This was the first step to define a gradient of activation going from the maximum projected value of CD69 (maximum activation) to the maximum projected value of TIM3 (maximum exhaustion) (**B**). (**C**) Z-scores of the original markers over PC2 and PC3. (**D**) Visual representation of the inter- and intra-patient heterogeneity, that shows how most of the patients present a relative homogeneous activation status of the Tcy. Each core is assigned an activation status ('Active', 'Transition', or 'Exhausted'). The cores are grouped for each patient, giving an at-a-glance representation of the heterogeneity of the activation status in different areas of the melanoma in the same patient. (**E**) The survival analysis in our data set shows a higher overall survival for brisk patients (left) and for patients with high levels of

*Figure 2 continued on next page*

*Figure 2 continued*

activation (right). Most importantly, the functional definition of activation/exhaustion shows improved prognostic value when compared to the brisk morphological parameter (p.value = 0.075 vs p.value = 0.31 log-rank test). The online version of this article includes the following figure supplement(s) for figure 2:

**Figure supplement 1.** Validation of the prognostic implications of our activation score in an independent cohort.
**Figure supplement 2.** qPCR and shotgun proteomics.
**Figure supplement 3.** Definition of activation.
**Figure supplement 4.** Definition of Activation in the Core Level (**A**) and Patient Level (**B**).
**Figure supplement 5.** Biplot showing the projection of the Tcy cells over PCs 2 and 3.

further analysis. Cell phenotypes were further clustered into functional groups using a set of functional markers. The functional clustering resulted in a total number of 47 functional cell populations (*Figure 4A*).

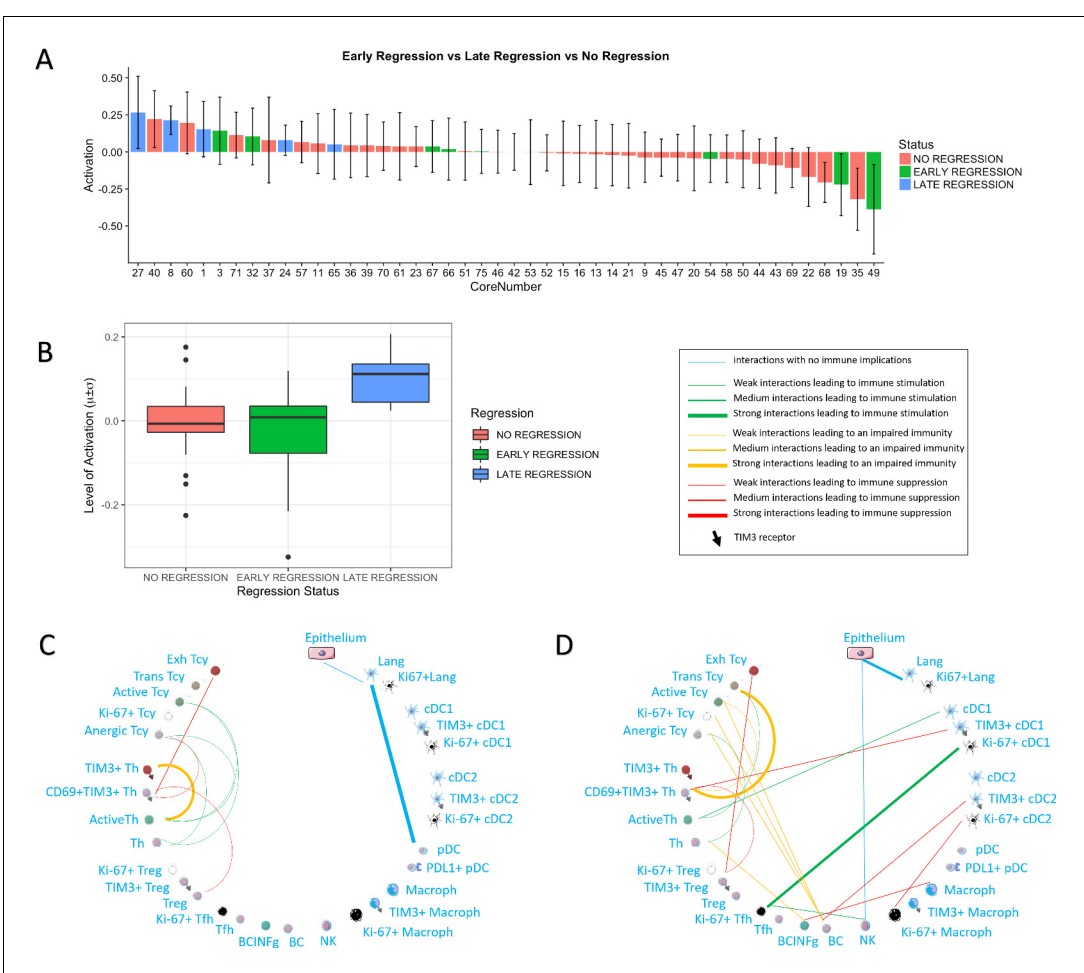

**Figure 3.** Activation status and neighborhood analysis in late, early and no regression. (**A**) The histogram shows the distribution of the different cores according to the activation levels of the Tcy. The color code identifies the presence and the type of regression areas in the cores. The cases with late regression are all in the left part of the histogram, showing higher levels of activation compared to cores with early regression or without regression. (**B**) This can be visualized also as a box plot. The neighborhood analysis for late (**C**) and early (**D**) regression shows an enrichment in immune-stimulating interactions in the first and more interactions leading to immune impairment in the latter. The thickness of the edge in the network represents the level of interaction between the different cell types. The colour of the line indicates interactions leading to immune suppression (red), to immune stimulation (green), to a probably sub-optimal/impaired immune stimulation (orange), no immune implications (blue).

**Table 1.** Statistical analysis.

Histopathological parameters were correlated with core status (Active/Transition/Exhausted) using pairwise t-tests with pooled standard deviation. Several histopathological parameters were correlated to the average level of activation of each core.

| Parameter | Comparison | Statistical test | Multiple testing correction | p value |
|---|---|---|---|---|
| Brisk Infiltration | Brisk vs Non Brisk | t test | No | 0.8453 |
| Brisk Infiltration | Brisk vs Brisk In Non Brisk | pairwise t test | Yes | 1 |
| Brisk Infiltration | Brisk vs Non Brisk In Non Brisk | pairwise t test | Yes | 1 |
| Brisk Infiltration | Brisk In Non Brisk vs Non Brisk In Non Brisk | pairwise t test | Yes | 1 |
| Regression | Regression vs No Regression | t test | No | 0.6275 |
| Regression | Early Regression vs No Regression | pairwise t test | Yes | 0.329 |
| Regression | Late Regression vs No Regression | pairwise t test | Yes | 0.031* |
| Regression | Late Regression vs Early Regression | pairwise t test | Yes | 0.022* |
| Count Lymphocytes | Number of Lymphocytes vs Level of Activation | linear model | No | 0.3714 |
| Histotype | LMM vs NMM | pairwise t test | Yes | 0.027* |
| Histotype | LMM vs SSMM | pairwise t test | Yes | 0.021* |
| Histotype | NMM vs SSMM | pairwise t test | Yes | 0.761 |
| Breslow Thickness | Breslow vs Level of Activation | linear model | No | 0.9883 |
| Ulceration | Positive vs Negative | t test | No | 0.7252 |
| Number of Mitoses | More than 6 vs 1 to 6 | pairwise t test | Yes | 1 |
| Number of Mitoses | More than 6 vs 0 | pairwise t test | Yes | 1 |
| Number of Mitoses | one to 6 vs 0 | pairwise t test | Yes | 1 |

The most abundant cell population consisted of melanoma cells (41.76%). Most of the melanoma cells expressed both melanocytic markers Melan A and S-100, while two minor groups of melanoma cells had loss of expression of one of the two markers. The second most abundant cell type were the macrophages ('Macroph', CD68+CD163+Lysozyme+HLA-DR+, 11.48%), one quarter of them expressing the immunosuppressive marker TIM3. Epithelial cells represented 8.4% of the population of our data set. The lymphoid compartment accounted for multiple subpopulations. Tcy (CD3+CD8+) and T helpers ('Th', CD3+CD4+FOXP3-) were the most abundant subtypes, accounting respectively for 11.19% and 10.10% of all the cells, while regulatory T cells ('Treg', CD3+CD4+FOXP3+) represented 2.82% of the cells. We interpreted the last T cell cluster as T follicular helpers ('Tfh', CXCL13+PD1+, 2.57%) even though this cluster did not express the full Tfh phenotype. Within Tcy, we could identify the active (CD69+OX40+/-, 16,1% of all the lymphocytes), transition (balanced expression of both exhaustion and expression markers, 29.6%) and exhausted (high expression of LAG3 and/or TIM3, low/absent CD69 and OX40, 12.8%) functional subgroups. Moreover, we found a clonally expanding subgroup (6%), and one with low or absent expression of all functional markers, that we defined as 'anergic' (34,9%) but that could represent also naive T cells. Th could also be further divided according to their expression of activation and exhaustion markers (28.5% ThCD69+, considered active, 18.46% ThCD69+TIM3+, considered transitional, and 13.08% ThTIM3+, considered immunosuppressive). NK cells, as expected in melanoma, were extremely infrequent (0.51%), and even more infrequent were the B cells ('BC', CD20+), that represented 0.12% of the total. The CD20 negative cells characterized by high expression of IRF4 and Blimp1 ('PC', 1,19%) were interpreted as plasma cells (*Caicedo et al., 2017*). Finally, we could identify among the dendritic cell

group the classical dendritic cells type 1 ('cDC1', CD141+CD4+IRF8+, 4.13%), classical dendritic cells type 2 ('cDC2', CD1c+CD4+HLA-DR+, 2.41%), Langerhans cells ('Lang', CD1a+Langerin+, 2.13%), and plasmacytoid dendritic cells ('pDC', CD123+, 0.33%). In both the two classical dendritic cell subgroups an immunosuppressive TIM3+ subpopulation was identifiable, while in the pDC group a small subpopulation was found to express PD-L1. No immunosuppressive subpopulation was identified among the Langerhans cells. Some subpopulations were statistically significantly different (p-val <0.05) comparing brisk vs non-brisk and active vs transition vs exhausted. Brisk cases were significantly enriched in BC (p-val = 0.041), TIM3+cDC1 (p-val = 0.024), macrophages (p-val = 0.043) (including the proliferating subgroup (p-val = 0.034)), NK (p-val = 0.011), anergic Tcy (p-val = 0.039) and proliferating Tcy (p-val = 0.006) (*Figure 4B*). Active cores had, together with more active Tcy (p-val <0.001), higher percentages of Tcy in transition (p-val = 0.030), transition cores more Th (p-val = 0.057), while exhausted cases had more TIM3+Tregs (p-val = 0.018) and more proliferating melanoma cells (p-val = 0.045) (*Figure 4C*).

## Neighborhood analysis

We applied neighborhood analysis in order to systematically identify social networks of cells and draw conclusions on actual cell-cell interactions. Macrophages and epithelial cells were in general most often located in strict proximity to the melanoma cells, without differences among the functional or morphologic categories. Brisk cases showed more Tcy in close proximity to melanoma cells than non-brisk cases, as expected (*Figure 5E*). Interestingly, brisk cases had a higher prevalence specifically of transition and active Tcy in contact with melanoma cells compared with non-brisk cases (Active/Exhausted ratio: Brisk = 2.108762, Non-Brisk = 1.331195), that instead had relatively more exhausted and anergic cells in contact with melanoma cells (Active/Anergic ratio: Brisk = 1.743081, NonBrisk = 0.7704947).

To understand what is the immune context that determines activation and exhaustion, we compared the results of neighbourhood analysis between active (*Figure 5A*) and exhausted (*Figure 5B*). The detection of the interaction of Langerhans cells with the epithelium, identifiable in both functional groups, was considered as positive control and proof of concept for the method. Some other interactions were present in both functional groups: TIM3+ macrophages and exhausted/transitional Tcy, CD69+TIM3+Th and transitional Tcy, active Th and anergic Tcy, TIM3+cDC1 and CD69+TIM3+Th. Cell-cell interactions that were specific for active cases included: active Th and active Tcy, TIM3+macrophages and CD69+TIM3+Th, NK and Tfh, Tfh and exhausted Tcy, and pDC and Langerhans. On the other hand, cell-cell interactions specific for exhausted cases included: CD69+TIM3+Th and anergic Tcy, anergic Tcy and active Th, anergic Tcy and BC expressing INFgamma, CD69+TIM3+Th and TIM3+cDC2, TIM3+macrophages and transition Tcy, NK and PD-L1+pDC, and the different subtypes of Tfh and cDC1. Considering the brisk and non-brisk classification, instead, we could first of all observe that the neighborhood analysis profile of the brisk cases was very similar to that of active cases. Moreover, non-brisk cases showed more cell-cell interactions linked with immune suppression. Nevertheless, both categories were not totally overlapping with the active and exhausted plots, but rather presented a mixture of immune-stimulating and immune-suppressive interactions present either in the active of exhausted plots (eg: CD69+TIM3+Th interaction with anergic Tcy, common between brisk and exhausted, or the active Th toward the active Tcy present both in the active and non-brisk group). This data confirmed us once again that the morphological categories are functionally heterogeneous (*Figure 5C,D*).

Finally, we also compared neighbourhood analysis between cores with early and late regression. In late regression, a network of activating interactions between active Tcy and active Th was Present. Few immune suppressing interactions could be observed, in particular between Treg-CD69+TIM3+Th-Exhausted Tcy (*Figure 3C*). In early regression, the interactions between active Th and active Tcy disappeared in this group, to leave space to aggregates of B cells located in strict proximity with anergic, proliferating and active T cells and probably stimulated by TIM3+cDC2, counterbalancing the effect of the immune stimulation between cDC1 and active Th (*Figure 3D*).

## qPCR and shotgun proteomics

Since our TMA is composed exclusively of primary melanomas, one may object that in a metastatic setting, cases with mainly active TILs may not be detected, maybe because the immune system of

the patient is failing in keeping control of the tumor. Or maybe, in the metastatic setting, to see a diffuse TILs infiltrate in a metastatic nodule would have been possibly correlated with real activation and not with exhaustion, and only morphological evaluation could be enough to evaluate the activation status of the TILs. Since immunotherapy is generally administered only if the patient develops metastasis (even if since very recently adjuvant immunotherapy is starting to be introduced in the clinic), to confirm the existence of the same functional subcategories in metastatic melanoma samples, we perform qPCR followed by proteomics on microdissected TILs, comparing melanoma metastasis that we classified as brisk (if diffusely infiltrated by TILs) and non-brisk (if only partially infiltrated by TILs), using an 'absent' case (metastasis without TILs) and a 'tumoral melanosis' (complete melanoma regression with persistence of melanin-loaded macrophages) case as controls. Furthermore, we could measure directly the levels of expression of IFNg, the best indicator of CD8+ activation, for which no suitable antibody exists for FFPE material. 4/8 melanomas with a brisk TILs pattern, and 3/7 with non-brisk TILs pattern proved active, confirming that we could subclassify from a functional point of view also metastatic lesions (*Figure 2—figure supplement 2*). To correlate the gene expression measured by qPCR with the actual protein expression, we performed a proteomic analysis on the microdissected material in three representative cases, that is one with high IFNg expression ('IFNg-high'), one with the high LAG3 and no IFNg expression ('LAG3-high'), and one with none of the four markers strongly expressed ('none'). The results showed a higher number of proteins identified in the 'IFNg-high' sample (324) compared to 'LAG3-high' (93) and 'none' (134), with almost two thirds of them (210/324) not shared with the other two samples and enriched in proteins involved in different inflammatory pathways (innate immunity, TNFR1 signalling pathway, FAS signalling pathway, T cell receptor and Fc-epsilon receptor signalling pathway), including the interferon gamma-mediated signaling pathway. As we minimized the difference in the number of microdissected cells in each sample, these observations could only be explained by a higher production of pro-inflammatory proteins in the 'IFNg-high' sample.

## Discussion

The fact that not all brisk infiltrates have a good prognosis (*Saltz et al., 2018*; *Thorsson et al., 2018*), that TILs populations can be very heterogeneous (*Bernsen et al., 2004*; *Pao et al., 2018*) and that anti-PD-1 immunotherapy was found to support functionally activated T cells (*Krieg et al., 2018*) urged a thoughtful investigation of the functional status of the TILs. Since the molecules that mediate exhaustion are expressed upon activation in order to prevent the hyperactivity of the immune system (*Wherry, 2011*), to investigate the activation status of lymphocytes the simultaneous expression of several molecules must be evaluated at a single-cell level and at once, as a panel (*Weixler et al., 2015*). Moreover, the assessment of the interactions between cells requires preservation of the tissue architecture. Our study is the first to assess up to 40 markers on tissue sections at single cell resolution without losing their spatial distribution. We obtained by high-dimensional in situ immunotyping a snapshot of the co-expression patterns of activation and inhibition markers in tissue sections. OX40 was strongly correlated with Ki-67 and PD-1, confirming its role in sustaining Tcy clonal expansion (*Huang et al., 2015*). We didn't instead find correlation between LAG3 and TIM3. This could be due to the fact that LAG3 is expressed very early and only transiently during T cell activation (*Andrews et al., 2017*). Persistent T cell activation with sustained expression of LAG3 together with other exhaustion markers (e.g. PD-L1) results in T cell dysfunction (*Andrews et al., 2017*). LAG3+Tregs have been shown to suppress DC maturation (*Spranger et al., 2016*).

We could observe that both brisk and non-brisk cases can harbour predominantly exhausted TILs or predominantly active TILs; therefore, with the morphological classification (brisk – non brisk – absent), a complementary functional classification (active – transitional - exhausted) co-exists. The importance of going beyond the morphologic classification of TILs was previously raised by others, who claimed that a functional analysis could help in the application of a better immunoscore for therapeutic prediction (*Pao et al., 2018*). Only a minority (15%) of patients presented with all active cores, and 30% of patients presented with at least one active core, whereas the great majority of the patients presented with only exhausted or transition areas at the moment in which the melanoma was removed. Since the percentage of patients that obtains a durable response with single agent checkpoint inhibition therapy (*Ribas and Flaherty, 2015*; *Ribas et al., 2016*; *Robert et al., 2015*) lies between 15% and 30%, and since *Krieg et al.*

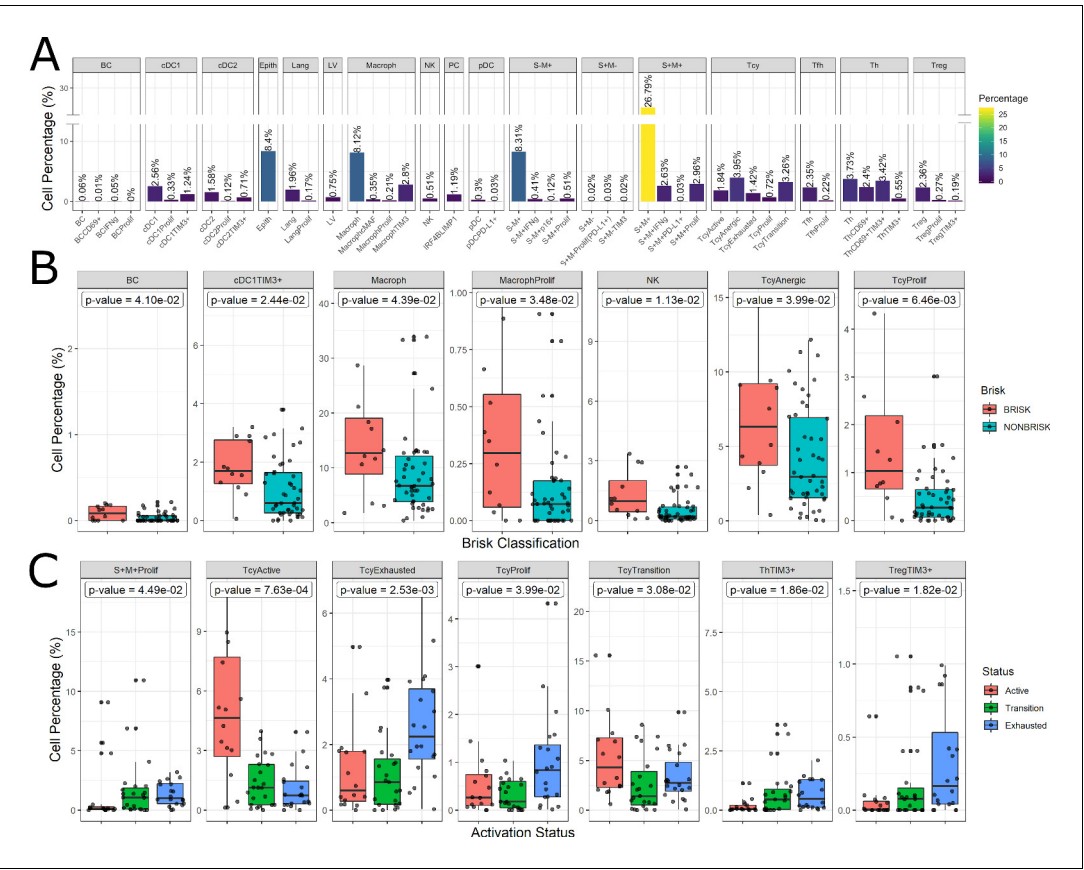

**Figure 4.** Distribution of the immune cells' subgroups and significant differences in the morphological and functional TILs categories. The percentage of cells for each inflammatory subpopulation across all the cores is showed in (**A**). The 17 phenotypic clusters are on the upper side of the graph, while at the bottom each of them is subdivided in the respective functional subclusters. BC = B cells; cDC1 = classical dendritic cells type 1; cDC2 = classical dendritic cells type 2; Epith = epithelial cells; PC = plasma cells; Lang = Langerhans cells; LV = lymph vessels; Macroph = macrophages; pDC = plasmocytoid dendritic cells; S-M+=S100+MelanA-melanoma cells; S+M-=S100-MelanA+=melanoma cells; S+M+=S100+MelanA+ melanoma cells; Tcy = cytotoxic T cells; Tfh = T follicular helpers; Th = T helpers; Treg = regulatory T cells; suffix: 'prolif'=proliferating, IFNg = interferon gamma. (**B**) Significant differences (p.value <0.05) in cell percentages between brisk and non-brisk categories (Wilcoxon rank sum test). (**C**) Significant differences (p.value <0.05) in cell percentages across the functional groups: Active, Transition, Exhausted (Kruskal-Wallis rank sum test).

The online version of this article includes the following figure supplement(s) for figure 4:

**Figure supplement 1.** Phenotype Identification.

*(2018)* reported that anti-PD-1 immunotherapy supports functionally activated T cells, it is tempting to speculate that the 'mostly active' TILs cases could correspond to the responders to immunotherapy, while the 'mostly exhausted' cases could benefit instead from combination approaches with immunotherapy and other type of therapies in order to rescue the exhausted T cells. Therefore, adding the functional evaluation could definitely improve the predictive value of the morphological TILs patterns in melanoma. Since spontaneous melanoma regression, present in only 10–35% of the melanomas, is considered to be the end-results of the melanoma-eliminating capacities of active TILs, we also studied the association of activation of TILs with early and late regression (*Botella-Estrada et al., 2014*). Our data showed a clear-cut association of activation of TILs with late regression areas, indirectly proving the functional meaning of an active infiltrate. We have to consider that the number of early (8) and late (5) regression areas is

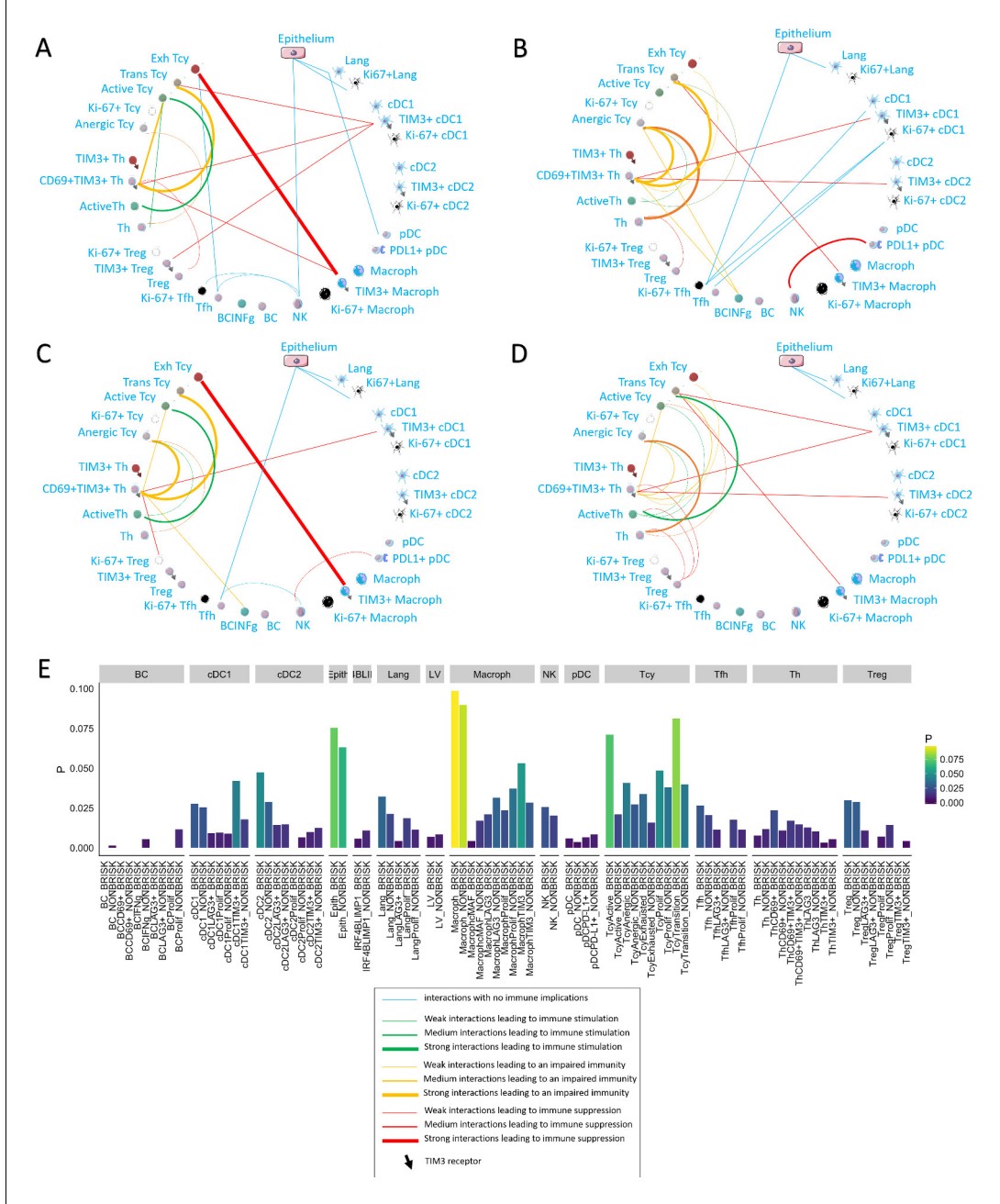

**Figure 5.** Neighborhood analysis in the morphological and functional TILs categories. The result of the neighborhood analysis for active (**A**), exhausted (**B**), brisk (**C**), and non-brisk (**D**) cases represented as cellular social networks. The thickness of the edge in the network represents the level of interaction between the different cell types. The colour of the line indicates interactions leading to immune suppression (red), to immune stimulation (green), to a probably sub-optimal/impaired immune stimulation (orange), no immune implications (blue). The brisk and active plots display a lot of similairies; nevertheless, the brisk and non brisk categories are not exactly overlapping with the active and exhausted plots, suggesting them to be the result of cases with activation and cases with exhaustion pooled together under the morphological labels. (**E**) The histogram shows the alternative approach of neighbourhood analysis tailored to explore specifically the interactions regarding melanoma cells. We observed that the main inflammatory cells subtypes in contact with melanoma cells are macrophages and (as expected) epithelial cells, both in brisk and non-brisk cases, followed by Tcy with active and in transition Tcy in brisk cases and proliferating and anergic Tcy in non-brisk cases. Other small differences between brisk and non-brisk cases are more TIM3+ cDC1, cDC2 and TIM3+ macrophages in contact with melanoma cells in brisk cases.

significantly smaller than the number of no regression areas (33). Thus, a larger cohort may be needed to validate these results. However, first of all, considering the sign of the reported differences (late regression more active than early regression and no regression) we do think that the reported results are coherent with the real biology. Early regression is a controversial entity among pathologists: in spite of the criteria mentioned above, there is no agreement about the fact that it should represent a real regression phase or rather a very pronounced brisk infiltrate. In this view, the dense inflammatory infiltrate in early regression may actually lead to late regression or may not, getting exhausted and disappearing without having actually a tangible antitumoral result. In agreement with this controversy, our data shows not only that early regression it is a very heterogeneous group displaying a great variability in activation levels (*Figure 3A*), but also that overall it has lower activation levels then late exhaustion (*Figure 3B*). Moreover, this data seems to be supported by neighborhood analysis. In late regression, a network of activating interactions between active Tcy and active Th was prevalent over few immune suppressing interactions (*Figure 3C*), while in early regression the dense infiltrate contains a more complex network of interactions with many of them contributing to immune impairment and, therefore, to the detected lower levels of activation (*Figure 3D*).

We obtained a functional picture of the inflammatory landscape in brisk versus non-brisk cases and in active/transition/exhausted TILs microenvironments. A previous study already casted some doubts on the significance of the brisk pattern, as they found no evidence of clonally expanded TILs in some cases with a brisk infiltrate (*Pao et al., 2018*), and hypothesized that clonally expanded T cells might represent not only cytotoxic cells but also regulatory cells. We indeed found not only immune stimulating cells such as proliferating and anergic Tcy, cDC1, and NK significantly increased in brisk cases, but also immune suppressive cells such as BC and macrophages. Active and exhausted cases were instead functionally coherent categories. An active microenvironment was enriched for active Tcy, while an exhausted microenvironment was enriched not only with Treg, possible origin of the exhaustion, but also with proliferating melanoma cells, possible effect of the lack of immune control over the neoplasia.

A slight increase in Treg between active and exhausted cases may not be able alone to justify the shift from an active to an exhausted microenvironment. To this end, adding the spatial information and making inferences about the interactions between the cells in the tissue using neighborhood analysis could represent a better way to investigate this dynamic. In this way, we could identify the most important differences that could explain the transition from an active to an exhausted environment. In active cases, it was possible to identify a stronger spatial association between active Th and active Tcy, while the same association was weaker in transition cases and disappeared in exhausted ones. In general, there was a decrease of interactions between the Th compartment and the active Tcy from active to exhausted cases. This could be explained observing the peculiar interactions with the other cell types in each of the functional states. In active cases, TIM3+ macrophages interacted with CD69+TIM3+ Th, as well as cDC1 expressing TIM3 interacted with TIM3+ Treg, possibly limiting their suppressive effect on the Tcy. In transition cases instead TIM3+ cDC1 are interacting only with CD69+TIM3+ Th and transitional Tcy, their block on Treg is removed and the Treg population is globally inhibiting all the Th subpopulation, probably reducing the strength of activation of the Tcy compartment. In fact, in active and exhausted cases the active Th are seen more often in contact with transitional Tcy, that are also the direct target of Tregs, while CD69+TIM3+ Th are in direct connection with exhausted Tcy. Starting from the transitional status, we see the appearance of interactions between BC expressing INFg-related molecules and CD69+TIM3+ Th and anergic Tcy, and between TIM3+ cDC2 and CD69+TIM3+ Th, confirming the immune suppressive role of this cell types. The presence of a shift of interaction of the active Th toward anergic Tcy observed in the exhausted cases may be a rescue mechanism, meant to induce new active Tcy starting from anergic/naïve Tcy but possibly contrasted by the effects of the surrounding immunosuppressive cells. Finally, in active cases, there is a strong interaction between TIM3+ macrophages and exhausted/transitional Tcy along all the statuses, confirming their prominent role in keeping the exhaustion in the tumoral microenvironment. Even if it may look like non-brisk cases have more activating interactions than brisk cases, the overall neighborhood analysis profile of brisk cases is very similar to the one of the active cases. Therefore, from our analysis emerges that the good prognosis of the brisk cases may be due to the fact that brisk cases have a profile of cell-cell interaction very similar to the one of active cases, witnessing that brisk cases and the fact that in brisk cases the melanoma cells are more

in contact specifically with active/transition Tcy than in non-brisk cases, that instead have more exhausted and anergic cells in contact with melanoma cells (as shown by the alternative neighborhood analysis method).

There are not many papers focusing on the activation status of the TILs in melanoma using a single-cell proteomics technique with spatial resolution. There are papers investigating the activation status of TILs at single-cell level in other tumors, such as non-small-cell lung cancer (*Guo et al., 2018*) or breast cancer (*Chung et al., 2017*). Though, these works use single-cell RNA sequencing, a non-spatial non-proteomics technique, to characterize exhaustion, losing the possibility to investigate the social network in which these exhausted cells are embedded. Some of these works show results that are similar to ours. For example, Guo et al. analyze, like us, treatment-naive patients and identifies a pre-exhausted T cell population that, similarly to our active/transition T cells, confer a better prognosis to patients when in bigger proportion in comparison to exhausted T cells. In the field of melanoma, *Tirosh et al. (2016)* gave a description of the immune landscape, limited to metastatic melanoma, and investigated the activation status of the T cells using single-cell RNAseq. In particular, they defined a 28-genes core exhaustion signature but also observed tumor-specific exhaustion signatures, hypothesizing that they could be the result of different previous treatments. This is indeed one of the limits of working with metastatic melanoma, not regarding instead our study, based on precedently untreated primary melanomas. Moreover, they identified T cells, B cells, macrophages, endothelial cells, and fibroblasts and assessed cell-cell interactions searching for genes expressed by a certain inflammatory cell type known to influence another cell type, finding that fibroblasts and macrophages expressed genes, mainly complement factors genes, correlated with T cell infiltration and with immune modulation of T cells. A similar approach within the same scope, that is to overcome the loss of information about the spatial distribution of the cells and infer cell-cell interactions, is presented by *Kumar et al. (2018)*. This work does not specifically deal with exhaustion but rather to a prediction for cell-cell communication based on scRNAseq-defined receptor/ligand expression. Despite the elegance of the approach, the mere presence of appropriate receptor/ligand pairs may not predict actual interaction in tissue; this is shown for example by the variety of the molecules used in a tissue-restricted fashion to phagocytize circulating particles by identical cell types (monocytes-macrophages) (*A-Gonzalez et al., 2017*). Talking about the limitations of their study, the authors themselves point out that "Several factors may lead to the identification of false positives [...] The level of transcripts does not necessarily correlate to protein expression for any gene [...] because scRNA-seq does not preserve spatial information, identified interactions in which the receptor and the ligand are membrane-bound may not occur when the corresponding cell types are not spatially co-localized in a tumor [...] approaches such as multiplexed immunofluorescence imaging or imaging mass cytometry can validate that membrane-bound interaction components are spatially co-localized. [...] Our methods provide a screening approach to identify potential ligand-receptor interactions that occur in a tumor microenvironment". As suggested here by the authors, our method have overcome these limitations, even though limited itself by the preselection of the markers included in our panel. Furthermore, the unique ability of the in situ analysis of cell interactions we can provide with our approach may yield a true representation of the network, producing unexpected results such as the mutual avoidance of receptor-ligand bearing T cells and macrophages in uterine leiomyosarcomas (*Manzoni et al., 2020*). Finally, a number of recent papers investigates the immune microenvironment focusing on response to therapy rather than prognosis and survival as main outcome. *Sade-Feldman et al. (2018)* identify two major CD8+ cell states, one with increased expression of genes linked to memory, activation and cell survival and one with enrichment in genes linked to cell exhaustion. A higher amount of memory/active cells at the baseline was found in responders to checkpoint inhibition, while exhausted CD8+ T cells were more abundant in non-responders. Also in their data set, TIM3, together with CD39, marked stronger than other genes the state of exhaustion of the CD8+ T cells. Active cytotoxic T cells as a hallmark for response to checkpoint inhibitor therapy were also found by *Riaz et al. (2017)*, whose work showed not only an increased number of TILs, NK and M1 macrophages in responders as a consequence of Nivolumab administration, but also enhancement of cytolytic pathways genes, possibly a bystander for cytotoxic T cell activation. Going beyond single-cell analyses, *Prat et al. (2017)*. uses a bulk digital mRNA expression method to define 12 signatures associated with response

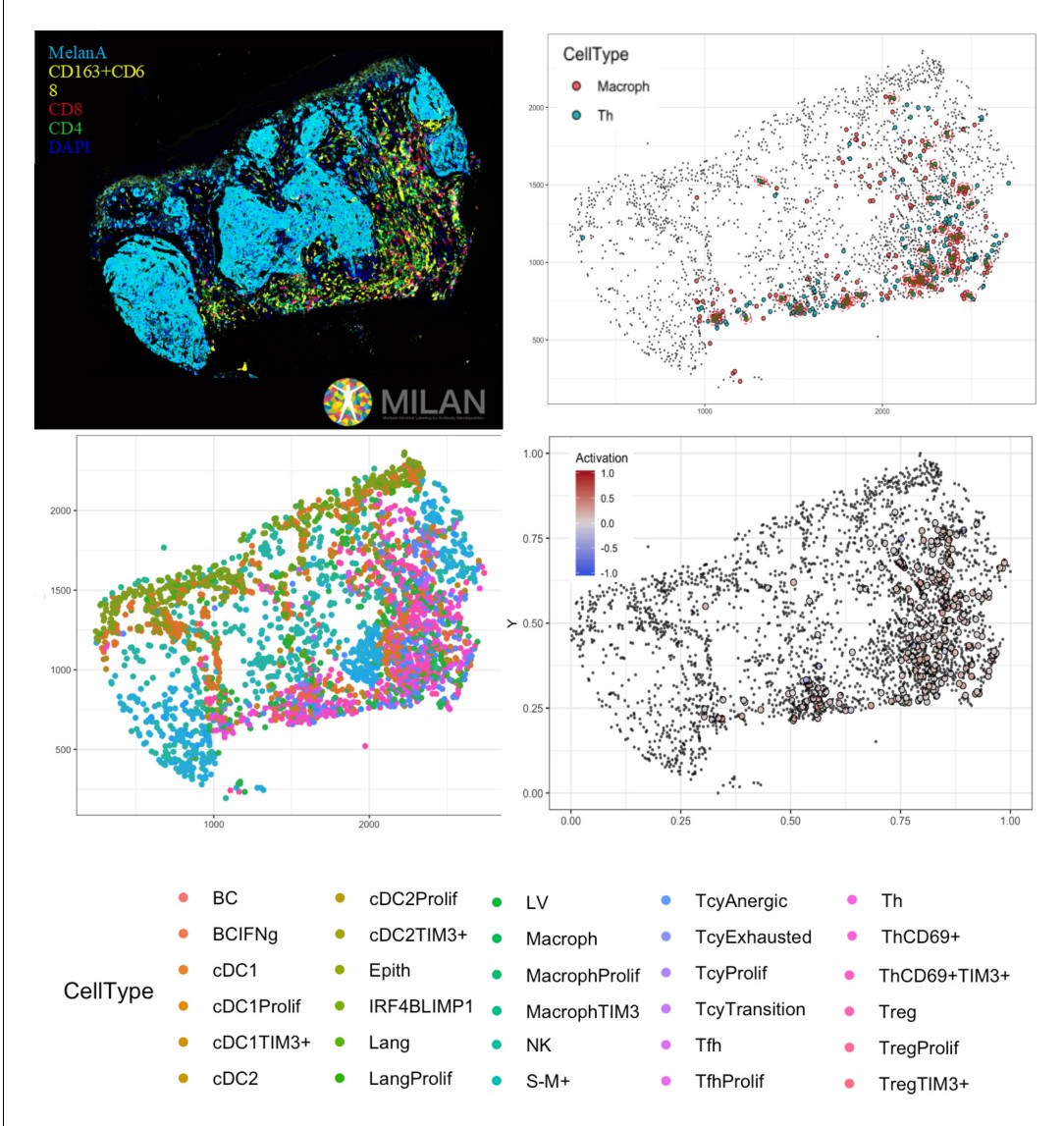

**Figure 6.** Complete digitalization of a core and practical example of a possible downstream analysis. Anti-clockwise: (1) images for the markers stained with the MILAN multiplexing technique (for this work 39 markers, but we reached in our laboratory an output of around 100 markers per single section) are acquired and composite images with some selected markers can be prepared. (2) All the markers are used to phenotypically identify all the cell subtypes present in the tissue and the tissue is digitally reconstructed. (3) Studies of the functional status of these cells can be done, for example we could localize exhausted and active cells in the tissue. (4) With neighborhood analysis, it is possible to identify the cells that are in proximity with each other more than chance can suggest, inferring possible interactions between these cell types (in the image, two cells identified as 'neighbors' are encircled in red).

and progression-free survival in different types of cancer, including melanoma. Among these signatures, T cell activation (both CD4 and CD8) and IFN activation are included. *Ayers et al. (2017)*. used the same bulk digital mRNA expression platform to develop a predictive 'interferon-gamma' gene signature, and observed that tumors with low signature scores characterized non-responders.

Because our melanoma dataset includes only primary melanoma collected before the era of immunotherapy (2005–2010) and only one patient was treated with immune-checkpoint based therapy, we can not directly correlate our T-cell activation signature with response to therapy and confirm these assumptions using our dataset. Moreover, even if we have highlighted with qPCR and shotgun proteomics that also metastasis can be classified as active and exhausted, it is not clear whether the tumor microenvironment of the primary melanoma is representative for the tumoral microenvironment once metastasized. We are currently prospectively collecting and investigating with the method proposed in this paper a cohort of immunotherapy-treated patients with paired biopsies (primary melanoma, pre-treatment and on-treatment samples at a later stage). A lot of predictive biomarkers for immunotherapy are now being proposed in literature, but it is becoming more and more clear that rather than reducing the number of parameters to obtain a prediction, a combination of different biomarkers will be needed to achieve the goal of personalized therapy. In the author's vision, the functional status of cytotoxic T cells (activation versus exhaustion) could be easily implemented in clinical practice using one of the fast multiplexing methods available on the market and a software that could automatize and speed up this type of analysis. The activation status of the TILs can be combined to other histopathological and immunological parameters (Breslow, TILs patterns, PD-L1 expression, spatial relationship of these exhausted and activated T cells to the tumor, …) and could therefore potentially play an important role in predicting overall survival and response to immune checkpoint therapy.

In conclusion, in this paper, we have shown that the activation status of the TILs does not necessarily parallel the morphological categories, and that within a single melanoma, the inflammatory response may vary considerably. The classification of primary melanomas based on the functional paradigm had an improved prognostic value when compared to the brisk classification. We hypothesize that the general good prognosis of melanomas with a brisk pattern of TILs could be based on the fact that the Tcy that are in contact with the melanoma cells at the moment that the melanoma is excised are still active, and consequently the melanoma is still under immune control. We have shown a bioinformatic pipeline that, starting from common immunofluorescence stainings, can transform the tissue into a digitized image, which represents the starting point for multiple deeper levels of analysis (*Figure 6*). In this study, we have also described the main interactions between the inflammatory subpopulations in an active, transition and exhausted environment, interactions that should be taken into consideration when assessing the response to immunotherapy and that will ultimately lead to the identification of functional inflammatory microenvironments that may benefit from personalized combined therapy protocols.

# Materials and methods

**Key resources table**

| Reagent type (species) or resource | Designation | Source or reference | Identifiers (RRID_AB) | Additional information (dilution) |
|---|---|---|---|---|
| Antibody | CD4 rabbit monoclonal EPR6855 | Abcam | | 1 µg/ml |
| Antibody | HLA-DR mouse monoclonal IgG2b SPM288 | Abcam | AB_1125217 | 1 µg/ml |
| Antibody | TAP2 mouse IgG1 monoclonal TAP2.17 | Abcam | | 1 µg/ml |
| Antibody | CD141 rabbit monoclonal EPR4051 | Abcam | AB_2201805 | 1 µg/ml |
| Antibody | MYC rabbit monoclonal EP121 | Sigma Aldrich | | 1 µg/ml |
| Antibody | FOXP3 mouse monoclonal IgG1 236A/E7 | Abcam | AB_445284 | 1 µg/ml |
| Antibody | MX1 Rabbit polyclonal | Abcam | AB_10678925 | 1 µg/ml |

*Continued on next page*

Continued

| Reagent type (species) or resource | Designation | Source or reference | Identifiers (RRID_AB) | Additional information (dilution) |
|---|---|---|---|---|
| Antibody | LAG3 mouse monoclonal IgG1 11E3 | Abcam | AB_776102 | 1 µg/ml |
| Antibody | PD-L1 rabbit monoclonal 28–8 | Abcam/Epitomics | AB_2687878 | 1 µg/ml |
| Antibody | CD1a rabbit monoclonal EP3622 | Abcam/Epitomics | AB_626957 | 1 µg/ml |
| Antibody | CD123 mouse monoclonal IgG2b NCL-L-CD123 | Leica-Microsystem/Novocastra | AB_10555271 | 1 µg/ml |
| Antibody | Phospho-Stat1 rabbit monoclonal 58D6 | Cell Signaling | AB_561284 | 1 µg/ml |
| Antibody | CD20 mouse monoclonal IgG2a L26 | Dako | AB_782024 | 1 µg/ml |
| Antibody | CD1a mouse monoclonal IgG1 O10 | Dako | | 1 µg/ml |
| Antibody | CD1c mouse monoclonal IgG1 2D4 | Dako | AB_2623049 | 1 µg/ml |
| Antibody | PRDM1 rat monoclonal 6D3 | Dako | AB_ | 1 µg/ml |
| Antibody | S100AB rabbit polyclonal | Dako | | 1 µg/ml |
| Antibody | CD56 mouse monoclonal IgG1 123C3.D5 | Neomarkers | AB_627127 | 1 µg/ml |
| Antibody | Ki-67 mouse monoclonal IgG2a UMAB107 | Origene | AB_2629145 | 2 µg/ml |
| Antibody | Lysozyme rabbit polyclonal | Origene | AB_1004766 | 1 µg/ml |
| Antibody | PD-1 mouse monoclonal IgG2a UMAB197 | Origene | AB_2629198 | 1 µg/ml |
| Antibody | TIM3 goat polyclonal | R and D | AB_355235 | 1 µg/ml |
| Antibody | CXCL13 mouse monoclonal IgG1 53610 | R and D | AB_2086049 | 1 µg/ml |
| Antibody | OX40 mouse monoclonal IgG1 Ber-ACT35 | Santa Cruz | AB_626897 | 1 µg/ml |
| Antibody | IRF4 goat monoclonal M-17 | Santa Cruz | AB_2127145 | 1 µg/ml |
| Antibody | cMAF rabbit monoclonal M-153 | Santa Cruz | AB_638562 | 1 µg/ml |
| Antibody | BCL6 rabbit monoclonal N3 | SCBT | AB_1158074 | 1 µg/ml |
| Antibody | CD16 mouse monoclonal IgG2a 2H7 | SCBT | AB_563508 | 1 µg/ml |
| Antibody | CD68 mouse monoclonal IgG3 PGM1 | Thermo Fisher | AB_10979558 | 1 µg/ml |
| Antibody | p16 mouse monoclonal IgG2a JC8 | SCBT | AB_785018 | 1 µg/ml |
| Antibody | MelanA rabbit monoclonal A19-P | NovusBio | AB_1987285 | 1 µg/ml |

*Continued*

| Reagent type (species) or resource | Designation | Source or reference | Identifiers (RRID_AB) | Additional information (dilution) |
|---|---|---|---|---|
| Antibody | Podoplanin rat monoclonal IgG2a NZ-1.2 | Sigma Aldrich | AB_10920577 | 1 µg/ml |
| Antibody | CD69 rabbit polyclonal | Sigma Aldrich | AB_2681157 | 1 µg/ml |
| Antibody | CD3 rabbit polyclonal | Sigma Aldrich | AB_2335677 | 1 µg/ml |
| Antibody | GBP1 rat monoclonal 4D10 | Sigma Aldrich | AB_828964 | 1 µg/ml |
| Antibody | Langerin rabbit polyclonal | Sigma Aldrich | AB_1078453 | 1 µg/ml |
| Antibody | IRF8 rabbit polyclonal | Sigma Aldrich | AB_1851904 | 1 µg/ml |
| Antibody | CD8 rabbit monoclonal SP16 | Thermo Fisher | AB_627211 | 1 µg/ml |
| Antibody | CD138 mouse monoclonal IgG1 MI-15 | Thermo Fisher | AB_10987019 | 1 µg/ml |

## Sample description

Twenty-nine invasive primary cutaneous melanomas from the Department of Pathology of the University Hospitals Leuven (KU Leuven), Belgium, were classified based on the H and E staining according to the pattern of the inflammatory infiltrate into brisk (six cases) and non-brisk (23 cases). According to their subtype, 24 superficial spreading melanomas, three nodular melanomas, and two lentigo maligna melanoma were included. Ethical approval was obtained from the Ethical Committee/IRB OG032 of the University Hospital of Leuven. After the approval, the study was identified with the number S57266. According to the Clinical Trial regalement no informed consent was needed due to the use of post-diagnostic left-over material. The patients' characteristics are resumed in *Supplementary file 6*. All the patients in our cohort were diagnosed between 2005 and 2010, before the era of immunotherapy (both in metastatic and adjuvant setting). Because only primary melanomas were included, all the patients received only surgery (broad excision) as first-line therapy. We did not find any significant differences between the 'Active' and 'Exhausted' groups according to age (t-test p.value = 0.6776382), gender (chisq-test p.value = 0.5814) or tumor location (chisq-test p.value = 0.1531). Given the relatively small number of patients included in the analysis and since none of the parameters seems to be related with patient activation, these factors were not accounted for the model in a multivariate analysis.

## TMA construction

Tissue Micro Arrays (TMAs) were constructed with the GALILEO CK4500 (Isenet Srl, Milan, Italy). For each patient, one to five representative regions of interest were sampled according to the size of the specimen and the morphological heterogeneity both of the melanoma and the infiltrate distribution. According to the morphological TILs pattern, we sampled brisk areas (i.e. six brisk areas in melanomas with brisk TILs pattern, termed 'brisk in brisk' and 10 brisk areas in melanomas with non-brisk TILs pattern, termed 'brisk in non-brisk') and non-brisk areas (i.e. 15 non-brisk areas in non-brisk melanomas, termed 'non-brisk in non-brisk'); in addition, areas showing 'early regression' (7)," late regression' (5) and 'no regression' (17) according to current morphological criteria (*Botella-Estrada et al., 2014*) were sampled. Eight morphological criteria were used to define regression: (1) Small or large areas with a decrease or an absence of melanoma cells in the dermal component of the tumor (2) Fibrosis (3) Inflammatory infiltrate (4) Melanophages (5) Neovascularization (7) Epidermal flattening (8) Colloid bodies (apoptosis of keratinocytes/melanocytes). Early regression was defined by small foci of melanoma disappearance (criterion1) and some fibrosis (criterion 2) but with a very dense inflammatory infiltrate (criterion 3) with any combination of the other criteria. Late regression was defined in presence of medium-to-large areas of melanoma disappearance substituted by very evident fibrosis (criterion 1 and 2) with any combination of the other criteria. After processing, cutting and staining of the TMA blocks, a total of 60 cores were available for analysis.

## Multiplex-stripping immunofluorescence

The multiplex staining was performed according to the MILAN protocol (*Cattoretti et al., 2001*) previously published (*Bolognesi et al., 2017*). It entails multiple rounds of indirect immunofluorescent staining, imaging and antibody removal via a detergent and a reducing agent. Unconjugated

**Table 2.** Multiplex antibody panel description.

| Protein | Concentration | Species | Clone | Company | RRID_AB: |
|---|---|---|---|---|---|
| CD4 | 1 µg/ml | rabbit Mab | EPR6855 | Abcam | N/A |
| HLA-DR | 1 µg/ml | mouse IgG2b | SPM288 | Abcam | 1125217 |
| TAP2 | 1 µg/ml | mouse IgG1 | TAP2.17 | Abcam | N/A |
| CD141 | 1 µg/ml | rabbit Mab | EPR4051 | Abcam | 2201805 |
| MYC | 1 µg/ml | rabbit Mab | EP121 | Sigma Aldrich | N/A |
| FOXP3 | 1 µg/ml | mouse IgG1 | 236A/E7 | Abcam | 445284 |
| MX1 | 1 µg/ml | Rabbit | | Abcam | 10678925 |
| LAG3 | 1 µg/ml | mouse IgG1 | 11E3 | Abcam | 776102 |
| PD-L1 | 1 µg/ml | rabbit Mab | 28–8 | Abcam/Epitomics | 2687878 |
| CD1a | 1 µg/ml | Rb mAb | EP3622 | Abcam/Epitomics | 626957 |
| CD123 | 1 µg/ml | mouse IgG2b | NCL-L-CD123 | Leica-Microystem/Novocastra | 10555271 |
| Phospho-Stat1 | 1 µg/ml | rabbit Mab | 58D6 | Cell Signaling | 561284 |
| CD20 | 1 µg/ml | mouse IgG2a | L26 | Dako | 782024 |
| CD1a | 1 µg/ml | mouse IgG1 | O10 | Dako | N/A |
| CD1c | 1 µg/ml | mouse IgG1 | 2D4 | Dako | 2623049 |
| PRDM1 | 1 µg/ml | rat | 6D3 | Dako | 628168 |
| S100AB | 1 µg/ml | rabbit | | Dako | N/A |
| CD56 | 1 µg/ml | mouse IgG1 | 123C3.D5 | Neomarkers | 627127 |
| Ki-67 | 2 µg/ml | mouse IgG2a | UMAB107 | Origene | 2629145 |
| Lysozyme | 1 µg/ml | rabbit | | Origene | 1004766 |
| PD-1 | 1 µg/ml | mouse IgG2a | UMAB197 | Origene | 2629198 |
| TIM3 | 1 µg/ml | goat | | R and D | 355235 |
| CXCL13 | 1 µg/ml | mouse IgG1 | 53610 | R and D | 2086049 |
| OX40 | 1 µg/ml | mouse IgG1 | Ber-ACT35 | Santa Cruz | 626897 |
| IRF4 | 1 µg/ml | goat | M-17 | Santa Cruz | 2127145 |
| cMAF | 1 µg/ml | rabbit | M-153 | Santa Cruz | 638562 |
| BCL6 | 1 µg/ml | rabbit | N3 | SCBT | 1158074 |
| CD16 | 1 µg/ml | mouse IgG2a | 2H7 | SCBT | 563508 |
| CD68 | 1 µg/ml | mouse IgG3 | PGM1 | Thermo Fisher | 10979558 |
| p16 | 1 µg/ml | mouse IgG2a | JC8 | SCBT | 785018 |
| MelanA | 1 µg/ml | rabbit Mab | A19-P | NovusBio | 1987285 |
| podoplanin | 1 µg/ml | rat IgG2a | NZ-1.2 | Sigma Aldrich | 10920577 |
| CD69 | 1 µg/ml | rabbit | | Sigma Aldrich | 2681157 |
| CD3 | 1 µg/ml | rabbit | | Sigma Aldrich | 2335677 |
| GBP1 | 1 µg/ml | rat | 4D10 | Sigma Aldrich | 828964 |
| Langerin | 1 µg/ml | rabbit | | Sigma Aldrich | 1078453 |
| IRF8 | 1 µg/ml | rabbit | | Sigma Aldrich | 1851904 |
| CD8 | 1 µg/ml | rabbit Mab | SP16 | Thermo Fisher | 627211 |
| CD138 | 1 µg/ml | mouse IgG1 | MI-15 | Thermo Fisher | 10987019 |

primary antibodies (see *Table 2*) are used, without the need of prioritizing any specific stain, there is no cell or tissue loss over >30 staining and stripping cycles (Manzoni M, et al. The adaptive and innate immune cell landscape of uterine leiomyosarcomas. Scientific Report, submitted), stained sections can be stored for additional subsequent experiments. In our previous publication, it is shown that the variation for repeated staining on different sections averages 3.1% (range 0–12.3%). Repeated staining while multiplexing (10 rounds) is usually less than 15% with some exceptions (e.g keratin 19). The range of variation after 30 cycles is even less than 15% and, most important, no new pixels are added or lost. In addition, we have found that only about 2–3% of all antibodies used are made unreactive just after a single stripping session. In conclusion, we believe that the MILAN protocol allows to safely detect most antigens surviving routine processing, even after 30 cycles, with variation in intensity within the variance of the technique itself.

## Image pre-processing

Fiji/ImageJ (version 1.51 u) were used to pre-process the images (File format: from. ndpi to.tif 8/16 bit, grayscale). Registration was done through the Turboreg and MultiStackReg plugins, by aligning the DAPI channels of different rounds of staining, saving the coordinates of the registration as Landmarks and applying the landmarks of the transformation to the other channels. Registration was followed by autofluorescence subtraction (Image process → subtract), previously acquired in a dedicated channel, for FITC, TRITc and Pacific Orange. A macro was written in Fiji/ImageJ and used for the TMA segmentation into single images. Cell segmentation, mask creation, and single-cell measurements were done with a custom pipeline using CellProfiler (version 2014-07-23T17:45:00 6c2d896). Quality Control (QC) over the Mean Fluorescence Intensity (MFI) values was performed using feature and sample selection. In short, those cells that did not have expression in at least three markers, and those markers that were not expressed in at least 1% of the samples were removed. MFIs were further normalized to Z-scores as recommended in Caicedo JC, et al (*Caicedo et al., 2017*). Z-scores were trimmed between −5 and +5 to avoid a strong influence of any possible outliers in the downstream analysis. The correlation between the different markers was calculated using Pearson's correlation coefficient.

## Functional analysis of TILs

We selected two activation (CD69, OX40) and two exhaustion (TIM3, LAG3) markers after literature review and preliminary testing of the antibody performance on control FFPE under the conditions of the multiplex protocol. The expression levels of these markers were measured selectively on CD8+ lymphocytes using a first mask focused only on CD8+ cells. Principal Component Analysis (PCA) was applied over the expression values to evaluate the functional structure of the data and to assign an activation value in the [−1, 1] range to each cell (Supplementary Information 1, *Figure 2—figure supplement 3*, *Animation 1*, *Animation 2*). Briefly, principal components (PCs) 2 and 3 were used as the rotation matrix revealed that PC1 contained all the markers in the same direction (same sign). The point of maximum activation (Activation = 1), was defined where the projected value of CD69 (marker of activation) over PCs 2 and 3 was at the maximum while the point of maximum exhaustion (Activation = −1) where the projected value of TIM3 (marker of exhaustion) over PCs 2 and 3 was at the maximum (*Figure 2*). The gradient of transition was defined between the previously defined points and the centroid of the projected dataset. Pairwise t-tests with pooled standard deviation (sd) were used to find significant differences in the level of activation of the images regarding multiple histopathologic parameters (brisk/non-brisk

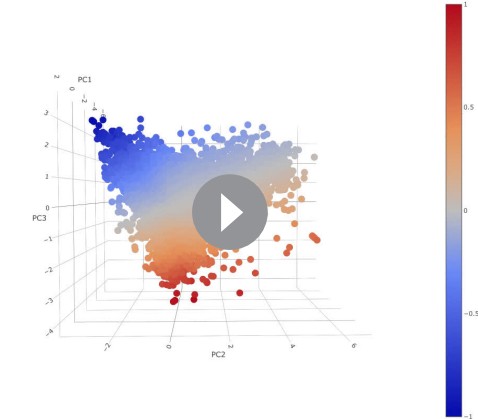

**Animation 1.** 3D scatter plot with PC1, PC2 and PC3. Animated in order to see the pyramidal shape of the 3D structure (rotating around the PC2 axis).
https://elifesciences.org/articles/53008#video1

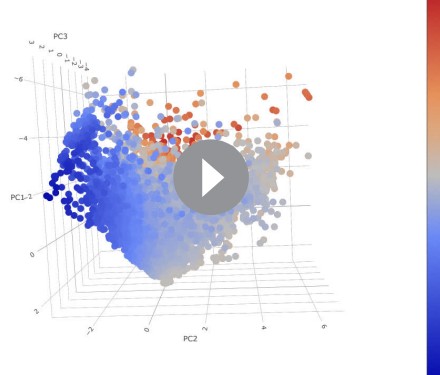

**Animation 2.** 3D scatter plot with PC1, PC2 and PC3. Animated in order to see the pyramidal shape of the 3D structure (rotating around the PC1 axis). https://elifesciences.org/articles/53008#video2

infiltrate, regression, number of lymphocytes, ulceration, Breslow thickness, mitoses, subtype of melanoma). For this purpose, the activation level of each image was represented by the mean of its cells while the intrinsic degree of heterogeneity was captured by the sd. Continuous values of the histopathological parameters were fitted using linear models instead. P-values were adjusted for multiple comparisons using the holm method. A cut-off of 0.05 was used as significance threshold for the adjusted p-values. Cores were further classified into: 'Active', 'Transition', and 'Exhausted' (from now on, indicating the core status) using one-tailed t-tests comparing the distribution of the activation values in a specific image versus the background distribution (combination of all images). P-values were adjusted using the False Discovery Rate (FDR) method. A cut-off value of 0.001 over the adjusted p-values was used as classification threshold. In order to obtain patient-specific read-outs, instead of combining scores from multiple cores, we pooled together all the cells from all the cores for each patient, and repeated the same that we did for the cores (to compare the distribution of cells for a specific patient against the background distribution). This classified each patient into one of the three categories ('Active', 'Transition', and 'Exhausted') (*Figure 2—figure supplement 4*). In order to evaluate the robustness of our patient classifications, we reclassified each patient 100 times using different sampling sizes (from 10 to 1000 cells, every 10). This resampling analysis confirmed that our classification of patients into 'Active'/"Exhausted' is very robust, requiring a relatively small number of cells to classify each patient reliably. We evaluated for every patient the minimum number of cells required to obtain the same significant classification in at least 95% of the simulations. The average value across all patients is 180 cells for active cases, and 174 for exhausted cases. Some cases, (patients 2, 3, 8, 12, 15, 18, and 21) required as little as 30 cells to obtain the same significant classification in at least 95% of the simulations. Only three cases (7, 9, and 17) required a relatively large number of cells (>360,~2 times the average) to obtain a significant classification in at least 95% of the simulations. We acknowledge that some patients classified as 'Transition' could have been identified as 'Active'/"Exhausted' would the number of identified Tcells had been bigger. For the survival analysis, the functional status of each patient was represented by the average level of activation of its cells and dichotomized into active (average level of activation >0) and exhausted (average level of activation < = 0). In order to validate the survival results from *Figure 2E* in another dataset, we obtained clinical and bulk RNA sequencing (RNA-seq) data from The Cancer Genome Atlas Skin Cutaneous Melanoma (TCGA-SKCM) dataset using the TCGA2STAT R package (https://academic.oup.com/bioinformatics/article/32/6/952/1744407). This dataset includes 460 patients from stages I-IV with 20501 genes annotated (HGNC format). However, our melanoma dataset includes only primary melanomas: most of the patients are stage I or II according to the 8th edition of the AJCC melanoma staging system. There is only one stage IIIC due to in-transit metastases at diagnosis and one stage IV due to primary metastases. Therefore, from the TCGA-SKCM dataset we included only those patients belonging to stages I and II for further analysis. For the deconvolution of bulk RNA-seq data, we used Cibersort (*Chen et al., 2018*). Cibersort offers a 22 immune cell gene signature (LM22) that includes CD8+ T cells but does not separate them into 'Active' and 'Exhausted'. Therefore, in order to build these gene signatures, we first obtained the single-cell RNAseq data from Tirosh et al (https://science.sciencemag.org/content/352/6282/189/tab-figures-data) present in the Gene Expression Omnibus (GEO) dataset (https://www.ncbi.nlm.nih.gov/geo/query/acc.cgi?acc=GSE72056). This dataset included 4645 cells from 19 tumors, each annotated with 23684 gene IDs (HGNC format). 1389 tumoral cells were removed leading to a dataset composed by 3256 cells with the following annotation: Bcells (512), CAF (56), Endothelial (62), Macrophages (119), NK (51), Tcells (2040), and unknown

(416). For every cell, we first calculated a distance value (Spearman correlation) to each gene signature from the ones included in the LM22 dataset from Cibersort. For each cell, the highest correlation was selected as the predicted cell type. Cells with a maximum correlation coefficient smaller than 0.25 (764) were removed from further analysis, leaving a total of 2492. *Supplementary file 1* shows the contingency matrix between the cell label given by Tirosh and the predicted cell type using Cibersort's profiles. Based on these profiles, we identified 908 CD8+ T cells in Tirosh dataset. We isolated those cells and ran our cell activation level analysis splitting them into 496 'Active' (Activation >0) and 412 'Exhausted' (Activation < = 0) CD8+ Tcells. In the proteomic level, our activation score used the expression of CD69, OX40, LAG3, and TIM3 markers on CD8 positive (CD8+) cells. The transcription factors encoding for these proteins found in the TCGA-SKCM dataset are summarised in *Supplementary file 2*. Then we built a 'T.cells.CD8.Active' and 'T.cells.CD8.Exhausted' profile using the average expression level of all the cells identified as Active/Exhausted using only the six genes included in *Supplementary file 2* obtaining the profiles specified in *Supplementary file 3*. We ran Cibersort in the TCGA-SKCM dataset, using the expression profiles in Supplementary File X3 and obtained the relative number of Active and Exhausted CD8+ T cells in each patient (*Supplementary file 4*). We labeled a patient as Active if the percentage of Active CD8+ T cells was bigger or equal to the number of Exhausted CD8+ T cells and exhausted otherwise.

| Predicted cell type | B cells | Cancer Associated Fibroblasts | Macrophages | Natural Killer | T cells | Unknown |
|---|---|---|---|---|---|---|
| B cells memory | 262 | 0 | 0 | 0 | 0 | 36 |
| B cells naive | 117 | 0 | 0 | 0 | 0 | 8 |
| Dendritic cells activated | 0 | 0 | 0 | 0 | 0 | 2 |
| Dendritic cells resting | 0 | 0 | 1 | 0 | 0 | 1 |
| Eosinophils | 0 | 0 | 0 | 0 | 0 | 1 |
| Macrophages M0 | 0 | 0 | 4 | 0 | 0 | 1 |
| Macrophages M1 | 0 | 0 | 6 | 0 | 0 | 0 |
| Macrophages M2 | 0 | 0 | 43 | 0 | 0 | 3 |
| Mast cells activated | 0 | 0 | 0 | 0 | 0 | 1 |
| Mast cells resting | 0 | 0 | 0 | 0 | 0 | 1 |
| Monocytes | 0 | 0 | 50 | 0 | 0 | 11 |
| Neutrophils | 0 | 0 | 0 | 0 | 0 | 3 |
| NK cells activated | 1 | 0 | 0 | 33 | 14 | 19 |
| NK cells resting | 1 | 0 | 0 | 15 | 2 | 5 |
| Plasma cells | 1 | 0 | 0 | 0 | 0 | 1 |
| T cells CD4 memory activated | 0 | 0 | 0 | 0 | 113 | 3 |
| T cells CD4 memory resting | 0 | 0 | 0 | 0 | 405 | 24 |
| T cells CD4 naive | 0 | 0 | 0 | 0 | 70 | 28 |
| T cells CD8 | 0 | 0 | 0 | 2 | 869 | 37 |
| T cells follicular helper | 0 | 1 | 0 | 0 | 267 | 16 |
| T cells gamma delta | 0 | 0 | 0 | 0 | 2 | 1 |
| T cells regulatory Tregs | 0 | 0 | 0 | 0 | 11 | 0 |

## Phenotypic identification

To evaluate the cell subpopulations, a second mask based on the DAPI nuclear staining contour expanded by five pixels was created. A two-tier approach was followed for the identification of cell subpopulations: phenotypic and functional. The phenotypic identification was conducted by applying three different clustering methods: PhenoGraph, ClusterX, and K-means, over the phenotypic markers: CD3, CD20, CD4, HLA-DR, Bcl6, CD16, CD68, CD56, CD141, CD1a, CD1c, Blimp1,

Langerin, Lysozyme, Podoplanin, FOXP3, S100AB, IRF4, IRF8, CD1a, CXCL13, CD8, CD138, CD123, PD-1, and MelanA. PhenoGraph and ClusterX were implemented using the cytofkit package from R (*Chen et al., 2016*). Clusters were represented by a vector containing the mean of each marker and were used to further associate them to a cell subpopulation using prior knowledge. For a phenotype to be assigned to a cell, at least two clustering methods should agree on their predicted phenotype. Prior to functional identification, PCA was repeated over the Tcy cells (CD8+) using CD69, OX40, LAG3, and TIM3 markers in order to confirm that with the new mask, the same dataset with the same structure as with the CD8+ mask could be retrieved (*Figure 2—figure supplement 5*). The functional identification was conducted by applying PhenoGraph over the functional markers: CD69, Ki-67, TAP2, GBP1, MYC, p16, MX1, OX40, c-Maf, PD-L1, LAG3, TIM3, and Phospho-Stat1 (with the exception of Tcy cells for which we used a personalized panel consisting of: CD8, CD69, OX40, LAG3, TIM3, PD-1, and Ki-67). Clusters were represented, associated to cell subpopulations, and evaluated for stability as described for the phenotypic identification. Significant differences in the cellular composition of the cores based on activation status were identified using Kruskal-Wallis rank sum test. The same approach was repeated for the brisk infiltrate histopathologic parameter using Mann-Whitney test.

## Neighborhood analysis

An unbiased quantitative analysis of cell-cell interactions was performed using an adaptation of the algorithm described in *Schapiro et al. (2017)* for neighbourhood analysis to systematically identify social networks of cells and to better understand the tissue microenvironment. Our adaptation also uses a kernel-based approach (radius = 30 px) to define the neighbourhood of a cell and a permutation test (N = 1000) to compare the number of neighbouring cells of each phenotype in a given image to the randomized case. This allows the assignment of a significance value to a cell-cell interaction representative of the spatial organization of the cells. Significance values were further classified into avoidance ($-1$), non-significant (0), and proximity (1) using a significance threshold of 0.001 (more significant that all the random cases). Interactions across images were integrated according to *Equation 1*:

$$P_{i,j} = \frac{\sum_{k=1}^{M} \left( c_{i,j,k} \cdot \sqrt{N_{i,j,k}} \right)}{\sum_{k=1}^{M} \left( \sqrt{N_{i,j,k}} \right)} \tag{1}$$

where $C_{i,j,k}$ is the significance value ($-1$, 0, or 1) of the interaction between cell types $i$ and $j$ for image $k$, and $N_{i,j,k}$ is the geometric average of the number of cells of type $i$ and $j$ for image $k$. Cell-cell interactions were considered strong if they were significant in at least 75% of the N-adjusted cases (abs(P)>0.75), moderate if 50%, (0.5 < abs(P)<=0.75), weak if 25% (0.25 < abs(P)<=0.5), and non-significant otherwise (abs(P)<=0.25). A comparative analysis of the above described method was performed for the different core statuses as well as for the different brisk infiltrate cases (*Figure 5*).

Even though neighborhood analysis allows evaluation of cell-cell interactions, the mathematical model applied is limited in cases where there are dominant cell types that grow in nests, with cells packed next to each other. Therefore, for melanoma cells, we evaluated the closest neighbors by counting the number of cells of each subpopulation (apart from melanoma cells) that were in their neighborhood and divided the amount by the geometric average of the number of melanoma cells and the number of cells of the specific population across all the cores in which the specific population appears. This analysis was repeated for the different brisk and activation cases.

## Laser microdissection

18 fresh frozen melanoma metastases with different types of TILs patterns (nine brisk, seven non brisk, 1 absent and one tumoral melanosis) were collected in the Department of Pathology of the University Hospitals Leuven (KU Leuven), Belgium. 10 micrometre-thick sections were cut from each fresh frozen block and put on a film slide (Zeiss, Oberkochen, Germany). Sections were stained with crystal violet. Areas with dense TILs infiltrate were microdissected with the Leica DM6000 B laser microdissection device (Leica, Wetzlar, Germany). A calculation was made in order to microdissect the same surface in all the samples in order to minimize differences between the samples (around 10,000 lymphocytes/sample).

## qPCR of laser microdissected samples

RNA extraction was done with RNeasy Plus Micro Kit (Qiagen) according to the protocol. cDNA retrotranscription followed by an amplification step was done with Ovation Pico SL WTA System V2 (Nugen) according to protocol. Primers for Interferon gamma (INFg, forward 'TGTTACTGCCAGGACCCA' and reverse 'TTCTGTCACTCTCCTCTTTCCA'), TIM3 (forward 'CTACTACTTACAAGGTCCTCAGAA' and reverse 'TCCTGAGCACCACGTTG'), LAG3 (forward 'CACCTCCTGCTGTTTCTCA' and reverse 'TTGGTCGCCACTGTCTTC'), CD40-L (forward 'GAAGGTTGGACAAGATAGAAGATG' and reverse 'GGATAAGGATCTTTCTCCTGTGTT'), CD45 (forward 'GCTACTGGAAACCTGAAGTGA' and reverse 'CACAGATTTCCTGGTCTCCAT'), Beta2microglobulin (forward 'ACAGCCCAAGATAGTTAAGTG' and reverse 'ATCTTCAAACCTCCATGATGC'), HPRT (forward 'ATAAGCCAGACTTTGTTGGA' and reverse 'CTCAACTTGAACTCTCATCTTAGG') were designed with Perl Primer and tested in our laboratory. 96-wells plates were loaded with Fast SYBR Green Master Mix, the primers and the samples in the recommended proportions, and analysed with the 7900 HT Fast Real-Time PCR system (Applied Biosystems). The log fold change (logFC) of the expression values toward the expression value of CD45 were calculated. If the log(IFNg/CD45) was positive, the sample was classified as positive. On the other hand, exhaustion was defined by expression of LAG3 and/or TIM3 with lack of IFNg and CD40L expression.

## Shotgun proteomics of laser microdissected samples

The materials used for the shotgun proteomics analysis were: Trifluoroacetic acid, MS grade porcine trypsin, DTT (dithiothreitol), IAA (Iodoacetamide), ABC (Ammonium Bicarbonate), HPLC grade water, acetonitrile (ACN), were from Sigma-Aldrich (Sigma-Aldrich Chemie GmbH, Buchs, Switzerland). All solutions for Mass Spectrometry (MS) analysis were prepared using HPLC-grade. LCM collected material corresponding to about $10^4$ cells for each sample group was re-suspended in 90 μl of bidistilled water and immediately stored at −80˚C. For the bottom-up MS analysis, all the samples were processed and trypsinized. Briefly, thawed cells were submitted to a second lysis adding 60 μl of 0.25% w/v RapiGest surfactant (RG, Waters Corporation) in 125 mM ammonium bicarbonate (ABC) and sonicated for 10 min. Samples were then centrifuged at 14 000 × g for 10 min. About 140 μl of supernatants were collected, transferred in a new tube and quantified using bicinchoninic acid assay (Pierce -Thermo Fisher Scientific). After 5 min denaturation (95˚C), proteins were reduced with 50 mM DTT in 50 mM ABC at room temperature and alkylated with 100 mM IAA in 50 mM ABC (30 min incubation in dark). Digestion of samples was performed overnight at 37˚C using 2 μg of MS grade trypsin. RG surfactant were removed using an acid precipitation with TFA at a final concentration of 0.5% v/v. Samples were then spun down for 10 min at 14,000 x g and supernatants collected for MS analysis. Peptide mixtures were desalted and concentrated using Ziptip μ-C8 pipette tips (Millipore Corp, Bedford, MA). An equal volume of eluted digests was injected at least three times for each sample into Ultimate 3000 RSLCnano (ThermoScientific, Sunnyvale, CA) coupled online with Impact HD UHR-QqToF (Bruker Daltonics, Germany). In details, samples were concentrated onto a pre-column (Dionex, Acclaim PepMap 100 C18 cartridge, 300 μm) and then separated at 40˚C with a flow rate of 300 nL/min through a 50 cm nano-column (Dionex, ID 0.075 mm, Acclaim PepMap100, C18). A multi-step gradient of 4 hr ranging from 4% to 66% of 0.1% formic acid in 80% ACN in 200 min was applied' (*Chinello et al., 2019*). NanoBoosterCaptiveSpray ESI source (Bruker Daltonics) was directly connected to column out. Mass spectrometer was operated in data-dependent acquisition mode, using CID fragmentation assisted by N2 as collision gas setting acquisition parameters as already reported (*Liu et al., 2015*). Mass accuracy was assessed using a specific lock mass (1221.9906 m/z) and a calibration segment (10 mM sodium formate cluster solution) for each single run. Raw data from nLC ESI-MS/MS were elaborated through DataAnalysis v.4.1 Sp4 (Bruker Daltonics, Germany) and converted into peaklists. Resulting files were interrogated for protein identification through in-house Mascot search engine (version: 2.4.1), as described (*Liu et al., 2015*). Identity was accepted for proteins recognized by at least one unique and significant (p-value<0.05) peptide.

## Pathways analysis

Gene-set enrichment analysis was performed with DAVID 6.8 (*Huang et al., 2009*; *Dennis et al., 2003*). Pathways were visualized and partially analysed with STRING v10 (*Szklarczyk et al., 2015*).

## Statistical analysis

*Supplementary file 5* resumes the statistical tests chosen for the analysis with the justification for their choice.

## Acknowledgements

Francesca Maria Bosisio is funded by the MEL-PLEX research training programme ('Exploiting MELanoma disease comPLEXity to address European research training needs in translational cancer systems biology and cancer systems medicine', Grant agreement no: 642295, MSCA-ITN-2014-ETN, Project Horizon 2020, in the framework of the MARIE SKŁODOWSKA-CURIE ACTIONS).

Asier Antoranz is funded by the SyMBioSys research training programme ('Systematic Modeling of Biological Systems'), grant agreement no: 675585, MSCA-ITN-2015-ETN, Project Horizon 2020, in the framework of the MARIE SKŁODOWSKA-CURIE ACTIONS. Maddalena Maria Bolognesi is supported by a clinical research project BEL114054 (HGS1006-C1121) of the University of Milano-Bicocca and GlaxoSmithKline, UK. Thanks to Mario Faretta for providing essential software, AMICO.

## Additional information

### Competing interests

Asier Antoranz: Affiliated with ProtATonce Ltd. The author has no other competing interests to declare. Maddalena Maria Bolognesi: Has received funding from GlaxoSmithKline. The author has no other competing interests to declare. Leonidas Alexopoulos: Affiliated with ProtATonce Ltd. The other authors declare that no competing interests exist.

### Funding

| Funder | Grant reference number | Author |
| --- | --- | --- |
| Horizon 2020 Framework Programme | 642295 | Francesca Maria Bosisio |
| Horizon 2020 Framework Programme | 675585 | Asier Antoranz |
| University of Milano-Bicocca | BEL114054 HGS1006-C1121 | Maddalena Maria Bolognesi |

The funders had no role in study design, data collection and interpretation, or the decision to submit the work for publication.

### Author contributions

Francesca Maria Bosisio, Conceptualization, Data curation, Formal analysis, Investigation, Methodology; Asier Antoranz, Maddalena Maria Bolognesi, Data curation, Software, Formal analysis, Investigation, Visualization, Methodology; Yannick van Herck, Clizia Chinello, Fulvio Magni, Data curation, Formal analysis, Investigation, Methodology; Lukas Marcelis, Investigation; Jasper Wouters, Formal analysis, Supervision, Investigation, Methodology; Leonidas Alexopoulos, Funding acquisition, Methodology, Project administration; Marguerite Stas, Conceptualization, Data curation, Supervision, Funding acquisition, Project administration; Veerle Boecxstaens, Data curation, Investigation; Oliver Bechter, Data curation; Giorgio Cattoretti, Conceptualization, Data curation, Formal analysis, Supervision, Funding acquisition, Investigation, Methodology; Joost van den Oord, Conceptualization, Data curation, Formal analysis, Supervision, Funding acquisition, Investigation, Methodology, Project administration

### Author ORCIDs

Francesca Maria Bosisio (ID) https://orcid.org/0000-0002-8874-2003
Lukas Marcelis (ID) http://orcid.org/0000-0002-5446-1801
Jasper Wouters (ID) http://orcid.org/0000-0002-7129-2990
Giorgio Cattoretti (ID) http://orcid.org/0000-0003-3799-3221

## Ethics

Human subjects: Ethical approval was obtained from the Ethical Committee/IRB OG032 of the University Hospital of Leuven. After the approval, the study was identified with the number S57266. According to the Clinical Trial regalement no informed consent was needed due to the use of post-diagnostic left-over material.

### Decision letter and Author response

Decision letter https://doi.org/10.7554/eLife.53008.sa1
Author response https://doi.org/10.7554/eLife.53008.sa2

## Additional files

### Supplementary files

- Source code 1. Code for all the analyses included in the paper.
- Source data 1. Image metadata.
- Source data 2. Annotated single cell data and patient survival.
- Source data 3. Correlation histopathological data and lymphocyte features.
- Supplementary file 1. Contingency table between the cell labels given by *Tirosh et al. (2016)* and the predicted cell type from the LM22 signatures from Cibersort. The highest spearman correlation was obtained for the predictions.
- Supplementary file 2. Transcription factors selected to represent the proteins used to calculate the activation score.
- Supplementary file 3. Gene expression signatures for the T.cells.CD8.Active and T.cells.CD8. Exhausted cell types.
- Supplementary file 4. Relative percentages of T.cells.CD8.Active and T.cells.CD8.Exhausted cell types in the TCGA-SKCM dataset (stages I and II).
- Supplementary file 5. Justification of the statistical test chosen for each analysis.
- Supplementary file 6. Patient metadata.
- Transparent reporting form

### Data availability

All data generated or analysed during this study are included as source data files. Code for all the analyses included in the paper has been provided as Source code 1.

The following previously published dataset was used:

| Author(s) | Year | Dataset title | Dataset URL | Database and Identifier |
|---|---|---|---|---|
| Cancer Genome Atlas Research Network | 2008 | TCGA-SKCM | https://www.ncbi.nlm.nih.gov/projects/gap/cgi-bin/study.cgi?study_id=phs000178.v10.p8 | NCBI dbGaP, TCGSKCM phs000178 |

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

## Appendix 1

### Supplementary data

The followed pipeline is detailed below:

PC2 and PC3 are mapped into polar coordinates.

$$\rho = \sqrt{(PC_2)^2 + (PC_1)^2}$$

$\varphi = atan2(PC_3, PC_2)$ where PC2 and PC3 are calculated from the rotation matrix

PC2 = 0.0444 · CD69 + 0.7048 · OX40 + 0.4764 · LAG3 − 0.5236 · TIM3

PC3 = −0.7505 · CD69 + 0.3656 · OX40 + 0.1196 · LAG3 + 0.5372 · TIM3

The point of maximum activation (Activation = 1) was defined as the point where the projected value of CD69 in PCs 2 and 3 reaches a maximum (**Figure 2—figure supplement 3**, point A). The angle corresponding to the multi-valued inverse tangent of the rotation vectors of PC3 and PC2 (atan2(PC3, PC2)) ($\varphi 0$) is added to $\varphi$.

$$\varphi' = \varphi + \varphi 0$$

The point of maximum exhaustion (Activation = −1) was defined as the point where the projected value of TIM3 in PCs 2 and 3 reaches a maximum (**Figure 2—figure supplement 3**, point B).

The line of transition (Activation = 0) was defined as the bisector between the projected vectors of LAG3 and OX40 over PCs 2 and 3 (Supplementary Data **Figure 6**, Transition Line).

The four resulting areas (**Figure 2—figure supplement 1 and** to 4) do not cover the same range of $\varphi$. Each area was scaled so that it covers 90 degrees ($\pi/2$ rads).

Finally, the value of activation of each cell was calculated as:

Activation = ρ · cos(ϕ'') where ρ is the radius and $\varphi''$ the scaled angle.

