## [Decision Letter]

**Acceptance summary:**

We believe this work is highly valuable for the scientific community, especially the melanoma community, because it investigates and sheds light into the relationship between a tumour with true lymphocytic activation, and therefore with better prognosis, and the typical "brisk/non-brisk" morphological classification still in use by many research groups around the world. We also find these studies into the spatial relationship between cell types highly valuable, especially the confirmatory finding that brisk tumours have cytotoxic lymphocytes and melanoma cells in closer contact than non-brisk cases. This study makes a compelling case for considering both T cell activation and morphology when assessing patient prognosis, and therefore are delighted to accept it for publication.

**Decision letter after peer review:**

Thank you for submitting your article "Functional heterogeneity of lymphocytic patterns in primary melanoma dissected through single-cell multiplexing" for consideration by *eLife*. Your article has been reviewed by four peer reviewers, including C Daniela Robles-Espinoza as the Reviewing Editor and Reviewer #4, and the evaluation has been overseen by Tadatsugu Taniguchi as the Senior Editor.

The reviewers have discussed the reviews with one another and the Reviewing Editor has drafted this decision to help you prepare a revised submission.

Summary:

In this manuscript titled "Functional heterogeneity of lymphocytic patterns in primary melanoma dissected through single-cell multiplexing", Bosisio and collaborators describe a powerful method to infer the functional status of immune cells in melanoma FFPE slides through multiplex imaging and single-cell deconvolution. They analyzed 60 cores from 29 patients for a total of 179,304 identified cells forming 19 clusters of 47 functional cell populations. To put this into perspective, most single-cell sequencing studies consist of a maximum of tens of thousands and not hundreds of thousands of cells, so this represents a deep single-cell expression data set that can be used to address many interesting questions. In particular, the authors attempted to better refine the currently broad classification that is used to determine whether a tumor has an ongoing inflammatory reaction that would suggest the patient will respond well to immunotherapy.

Essential revisions:

1) In general, confounders and the role of heterogeneity are not well controlled in most of the analyses. It seems that it is rather a difficult task to score TIL status and functional states from any small piece of biopsy and assume it will accurately represent the whole tumor, or even multiple tumors a patient may have.a) Please clarify: How did the authors combine scores from multiple cores, how did they control for heterogenous core classifications, since the read-out is patient-specific (i.e., OS)? Treatments are not mentioned, but obviously if these tumors were collected within the last 8 years, these patients may have divergent treatment histories. How were these factored into the model, as well as age, gender, tumor location (especially with the potential confounding of immune cells in lymph-node metastases), etc?b) Reviewers suggest reproducing the survival analysis from Figure 2E in another dataset (could be one previously published and posted online) and see whether the conclusions replicate. It could perhaps be done with bulk RNA sequencing data, using deconvolution methods to search for the presence of the same markers.c) Related with the previous point, reviewers suggest looking more closely at the results of other single-cell melanoma studies that have investigated correlations with exhaustion (e.g., Kumar et al., 2018 doi.org/10.1016/j.celrep.2018.10.047) and putting in context results of the present study with those already published.

2) Neighbourhood analysis. The reviewers agree that this is one of the most interesting results in this manuscript as it adds information that other non-spatial single-cell-based are not able to provide.a) Please place relevant cell-cell interactions (e.g. melanoma:T-cell) in the main text instead of the supplementary materials, and mention panels C and D in the Results section.b) Please improve Figure quality by increasing font size and adding legends, and clarify why some cell types (e.g., TIM3 + cDC2) are not shown in Figure 4C.c) Please discuss how the fact that the "active" and "brisk" plots are nearly identical fits with the paper's main conclusion that the "brisk" state consists of both active and exhausted.

3) It is unclear what the qPCR and proteomics experiments add to the results. Please clarify how the fact that 50% of the brisk and 43% of the non-brisk tumours can be classified as "active" fits with the neighbourhood analysis, particularly point 2C above.

4) Please add more explanation in this manuscript about what the MILAN technique entails, given that it does not seem to be in widespread use.

5) Principal component analyses. Please clarify the following points:a) Are the PCAs in Figure 2 created with all markers or just the select markers shown in Figure 2?b) For the classification of cells into the "active" and "exhausted" states, were all cells from all patients put together in the analysis, and if so, did PC1 divide cells by patient? What variability did the axis of maximum variation capture?

6) Correlation between core status and tumour regression. Please answer the following questions:a) How is a late regression area defined?b) Do the authors think the reported correlation may have arisen by chance, given that No Regression vs Early regression (comparison of the most different states) did not show any differences? Were the differences that were detected in the expected direction (i.e. early regressed tumours had a higher activation status than late regressed tumours)?

7) The reviewers agreed that it is necessary to add a section in the discussion regarding how the authors' results compare and fit with previous publications, in particular Sade-Feldman et al., 2018; Ayers et al., 2017; Prat et al., 2017; Riaz et al., 2017 and Tirosh et al., 2017.

8) Regarding statistics in general. The reviewers suggest displaying p-values in all Figures where there have been statistical tests and justify the use of the chosen tests. (For example, reviewers mention that authors could have used logistic regression to analyze the relationship between their variables, as well as ANOVA instead of t-tests with Holm's or FDR corrections). Clarifications about when multi-testing correction was applied (or not) should be added.

[Editors' note: further revisions were suggested prior to acceptance, as described below.]

Thank you for resubmitting your work entitled "Functional heterogeneity of lymphocytic patterns in primary melanoma dissected through single-cell multiplexing" for further consideration by *eLife*. Your revised article has been evaluated by Tadatsugu Taniguchi (Senior Editor) and a Reviewing Editor.

The reviewers agree that the manuscript has been substantially improved but there are some minor remaining issues that need to be addressed before acceptance, as outlined below:

Could you please add some discussion on the following:

1) To assign an activation state to patients, the authors didn't combine states per core but per cell – Doesn't this depend on the number of cells that were successfully assessed? Reviewers would like to see an acknowledgment that perhaps sampling more/less cells per patient could have an influence in this classification.

2) Can the authors please expand a little on the logic behind classifying "non-brisk with exhaustion" as poor prognosis one whereas "brisk with exhaustion" is classified as good prognosis? How do the authors define the relationship between these two different classification methods? Is morphology then also important to take into account even when the single cell status is being considered? (And, was this the comparison that got the better p-value? Were the other possibilities (e.g. classifying brisk with exhaustion as poor prognosis) also considered?

The reviewers would also like to see clarification for one of the points in the rebuttal letter, please. One of them wrote, "The two explanations for categorising the "transition" patients into active or exhausted (p values 0.053 or 0.079) sound exactly the same to me. What is different? (However, I do not think this is mentioned in the main text)."

---

## [Author Response]

Essential revisions:1) In general, confounders and the role of heterogeneity are not well controlled in most of the analyses. It seems that it is rather a difficult task to score TIL status and functional states from any small piece of biopsy and assume it will accurately represent the whole tumor, or even multiple tumors a patient may have.

A small remark before proceeding to answer the more detailed questions: our TMAs are derived from primary melanomas. Diagnosis nowadays is made earlier than in the past, resulting in thinner melanomas, less material for pathologic examination and a limited tumoral area. In all the patients included in the TMA, the number of cores used in this study depends on the number of areas with inflammation present in the tumor. This corresponds to an extensive sampling of the melanoma microenvironment, something not so easily doable for other cancer types characterized by bigger tumoral masses (i.e. colon carcinoma), but certainly achievable in primary melanoma.

a) Please clarify: How did the authors combine scores from multiple cores, how did they control for heterogenous core classifications, since the read-out is patient-specific (i.e., OS)? Treatments are not mentioned, but obviously if these tumors were collected within the last 8 years, these patients may have divergent treatment histories. How were these factored into the model, as well as age, gender, tumor location (especially with the potential confounding of immune cells in lymph-node metastases), etc?

In order to obtain patient-specific read-outs, instead of combining scores from multiple cores, we pooled together all the cells belonging to each patient. After assessing the activation level of every CD8^+^ lymphocyte present in the TMA (subsection “Functional analysis of TILs”, and supplementary file 1), a label was assigned to each core (“Active”, “Transition”, and “Exhausted”) using one-tailed t-tests comparing the distribution of the activation values in the cells from a particular core versus the background distribution (combination of all cores) (Figure 2—figure supplement 4A). P-values were adjusted using the False Discovery Rate (FDR) method. A cut-off value of 0.001 over the adjusted p-values was used as classification threshold. Then, we pooled together all the cells from all the cores for each patient and repeated the same that we did for the cores (to compare the distribution of cells for a specific patient against the background distribution). This classified each patient into one of the three categories (“Active”, “Transition”, and “Exhausted”) (Figure 2—figure supplement 4B).

This explanation has been added to the text (Subsection “Functional analysis of TILs”) and one more supplementary Figure with the last two images has been added (Figure 2—figure supplement 4).

We then evaluated if these groups had an effect on survival and saw that indeed “Active” patients had a better prognosis than those in “Transition” which at the same time had a better prognosis than the “Exhausted” patients.

However, in order to make a direct comparison with the brisk and non-brisk categories (2 groups), we pushed the transition patients to either the active or exhausted group based on the average level of activation of all its cells. If the average level of activation was bigger than 0, the “Transition” patient was defined as “Active” while if the activation level was smaller than 0, the “Transition” patient was defined as “Exhausted”. This reduced the p-value (log-rank test) from 0.25 using the three groups to 0.053 using the two groups.

**Author response image 2. respfig2:** 

We finally compared this approach to one in which the activation level of each patient was simply represented by the average activation level of all its cells and categorized into “Active” and “Exhausted” if that activation level was bigger or smaller than 0. This “simpler” case changes the label of a single patient from “Active” to “Exhausted” and increases the p-value from 0.053 to 0.079.

**Author response image 3. respfig3:** 

Regardless of the approach used to label our patients in either “Active” and “Exhausted” or “Active”, “Transition”, and “Exhausted”, the biological insight depicted from this type of analysis is always the same: “Active” patients show a better prognosis than “Exhausted” ones. This classification, moreover, is better than that of the “Brisk” classification as the p-value reduces from 0.36 to 0.053. Moreover, as the contingency matrix shows, Brisk and Activation are not the same:

**Author response image 4. respfig4:** 

In fact, Brisk and NonBrisk groups are very well balanced in the “Active” and “Exhausted” groups.

**Author response image 5. respfig5:** 

If we combine both parameters (Brisk and Active), we can redefine as the bad prognosis group NonBrisk patients with exhaustion and the good prognosis patients all the other 3 groups (Brisk patients with activation, Brisk patient with Exhaustion, and NonBrisk patients with activation). The resulting KM in this case is the following:

**Author response image 6. respfig6:** 

These graphs and this explanation have been considered by the authors as intermediate passages in the survival analysis and left out of the manuscript not to excessively increase its length. We can include them if the reviewers consider it necessary.

Regarding patient treatment, all the patients in our cohort were diagnosed between 2005-2010, before the era of immunotherapy (both in metastatic and adjuvant setting). Therefore, the treatment options at that time were quite limited and after progression on standard chemotherapy (Cisplatinum – Dacarbazine) patients often were being included in clinical trials. Moreover, only patients diagnosed with melanoma metastases during the follow-up were eligible for systemic treatment. Because we are including only primary melanoma as mentioned earlier, all of these patients received only surgery (broad excision) as first-line therapy. In our cohort, only 2 patients actually received systemic therapy (other eligible patients either refused therapy or were in too bad condition for both chemotherapy or clinical trials), both first line chemotherapy. Because of this variability and low amount of patients, we did not find any significant differences between the active and exhausted groups regarding treatment. (Chisq-test p-value = 0.3679)

**Author response image 7. respfig7:** 

Regarding age, we did not find any significant differences between the “Active” and “Exhausted” groups (t-test p.value = 0.6776382):

**Author response image 8. respfig8:** 

Regarding gender, we did not find any significant differences between “Active” and “Exhausted” patients (chisq-test p.value = 0.5814):

**Author response image 9. respfig9:** 

Regarding tumor location, we did not find any significant differences between “Active” and “Exhausted” patients (chisq-test p.value = 0.1531):

**Author response image 10. respfig10:** 

Given the relatively small number of patients included in the analysis and since none of the parameters seems to be related with patient activation, these factors were not accounted for the model in a multivariate manner. We added this part to the subsection “Sample description”.

b) Reviewers suggest reproducing the survival analysis from Figure 2E in another dataset (could be one previously published and posted online) and see whether the conclusions replicate. It could perhaps be done with bulk RNA sequencing data, using deconvolution methods to search for the presence of the same markers.

For the deconvolution of bulk RNA-seq data we used Cibersort (Chen et al., 2019 doi: 10.1007/978-1-4939-7493-1_12). Cibersort offers a 22 immune cell gene signature (LM22) that includes CD8^+^ T cells but does not separate them into “Active” and “Exhausted”. Therefore, in order to build these gene signatures, we first obtained the single cell RNAseq data from Tirosh et al., 2016 (doi: 10.1126/science.aad0501) present in the Gene Expression Omnibus (GEO) dataset (https://www.ncbi.nlm.nih.gov/geo/query/acc.cgi?acc=GSE72056). This dataset included 4645 cells from 19 tumors, each annotated with 23684 gene IDs (HGNC format). 1389 tumoral cells were removed leading to a dataset composed by 3256 cells with the following annotation: Bcells (512), CAF (56), Endothelial (62), Macrophages (119), NK (51), Tcells (2040), and unknown (416).

For every cell, we first calculated a distance value (Spearman correlation) to each gene signature from the ones included in the LM22 dataset from Cibersort. For each cell, the highest correlation was selected as the predicted cell type. Cells with a maximum correlation coefficient smaller than 0.25 (764) were removed from further analysis, leaving a total of 2492. Supplementary file 1 shows the contingency matrix between the cell label given by Tirosh and the predicted cell type using Cibersort’s profiles.

Based on these profiles, we identified 908 CD8^+^ T cells in Tirosh dataset. We isolated those cells and ran our cell activation level analysis (Figure 2—figure supplement 1) splitting them into 496 “Active” (Activation > 0) and 412 “Exhausted” (Activation <= 0) CD8^+^ Tcells. In the proteomic level, our activation score used the expression of CD69, OX40, LAG3, and TIM3 markers on CD8 positive (CD8^+^) cells. The transcription factors encoding for these proteins found in the TCGA-SKCM dataset are summarised in Supplementary file 2.

**Author response image 11. respfig11:** 

Then, we built a “T.cells.CD8.Active” and “T.cells.CD8.Exhausted” profile using the average expression level of all the cells identified as Active/Exhausted using only the 6 genes included in Supplementary file 2 obtaining the profiles specified in Supplementary file 3.

In order to validate the survival results from Figure 2E in another dataset, we obtained clinical and bulk RNA sequencing (RNA-seq) data from The Cancer Genome Atlas Skin Cutaneous Melanoma (TCGA-SKCM) dataset using the TCGA2STAT R package (Wan, Allen and Liu, 2016: doi.org/10.1093/bioinformatics/btv677). This dataset includes 460 patients from stages I-IV with 20501 genes annotated (HGNC format). However, our melanoma dataset includes only primary melanomas: most of the patients are stage I or II according to the 8th edition of the AJCC melanoma staging system. There is only 1 stage IIIC due to in-transit metastases at diagnosis and 1 stage IV due to primary metastases. Therefore, from the TCGA-SKCM dataset we included only those patients belonging to stages I and II for further analysis.

We ran Cibersort in the TCGA-SKCM dataset, using the expression profiles in Supplementary file 3 and obtained the relative number of Active and Exhausted CD8^+^ T cells in each patient (Supplementary file 4). We labelled a patient as Active if the percentage of Active CD8^+^ T cells was bigger or equal to the number of Exhausted CD8^+^ T cells and exhausted otherwise. Survival analysis revealed that the Active group of patients had better prognosis than the exhausted group of patients (log-rank p-value = 0.0082) validating the results obtained with our dataset.

These methods and the results have been added in the paper (subsection “Functional analysis of the TILs” and Results section) and Figure 2—figure supplement 1 has been added.

c) Related with the previous point, reviewers suggest looking more closely at the results of other single-cell melanoma studies that have investigated correlations with exhaustion (e.g., Kumar et al. doi.org/10.1016/j.celrep.2018.10.047) and putting in context results of the present study with those already published.

(Note: The papers suggested in point 7 of this revision came across in the PubMed search while looking for more articles to complete point 1c, therefore we have discussed them here.)

There are not many papers focusing on the activation status of the TILs in melanoma using a single-cell proteomics technique with spatial resolution. There are papers investigating the activation status of TILs at single-cell level in other tumors, such as non-small-cell lung cancer (Guo et al., (2018)) or breast cancer (Chung et al., (2017)). Though, these works use single-cell RNA sequencing, a non-spatial non-proteomics technique, to characterize exhaustion, losing the possibility to investigate the social network in which these exhausted cells are embedded. Some of these works show results that are similar to ours. For example, Guo et al., analyze, like us, treatment-naive patients and identifies a pre-exhausted T cell population that, similarly to our active/transition T cells, confer a better prognosis to patients when in bigger proportion in comparison to exhausted T cells. In the field of melanoma, Tirosh et al., 2016) gave a description of the immune landscape, limited to metastatic melanoma, and investigated the activation status of the T cells using single-cell RNAseq. In particular, they defined a 28-genes core exhaustion signature but also observed tumor-specific exhaustion signatures, hypothesizing that they could be the result of different previous treatments. This is indeed one of the limits of working with metastatic melanoma, not regarding instead our study, based on precedently untreated primary melanomas. Moreover, they identified T cells, B cells, macrophages, endothelial cells, and fibroblasts and assessed cell-cell interactions searching for genes expressed by a certain inflammatory cell type known to influence another cell type, finding that fibroblasts and macrophages expressed genes, mainly complement factors genes, correlated with T cell infiltration and with immune modulation of T cells. A similar approach within the same scope, that is to overcome the loss of information about the spatial distribution of the cells and infer cell-cell interactions, is presented in the referenced paper by Kumar et al., 2018). This work does not specifically deal with exhaustion but rather to a prediction for cell-cell communication based on scRNAseq-defined receptor/ligand expression. Despite the elegance of the approach, the mere presence of appropriate receptor/ligand pairs may not predict actual interaction in tissue; this is shown e.g. by the variety of the molecules used in a tissue-restricted fashion to phagocytize circulating particles by identical cell types (monocytes-macrophages) (Gonzalez et al., 2017). Talking about the limitations of their study, the authors themselves point out that:

“Several factors may lead to the identification of false positives when using our approach to identify potential cell-cell interactions […] The level of transcripts does not necessarily correlate to protein expression for any gene […] because scRNA-seq does not preserve spatial information, identified interactions in which the receptor and the ligand are membrane-bound may not occur when the corresponding cell types are not spatially co-localized in a tumor […] approaches such as multiplexed immunofluorescence imaging or imaging mass cytometry can validate that membrane-bound interaction components are spatially co-localized. However, these approaches are not suitable for high-throughput screening of ligand-receptor interactions. In addition, studies examining cell-cell interaction in the tumor microenvironment should consider the specificity of those interactions to the tumor microenvironment. […] Our methods provide a screening approach to identify potential ligand-receptor interactions that occur in a tumor microenvironment.”.

As suggested here by the authors, our method have overcome these limitations: even though limited by the preselection of the markers, it can be used for validation of their screening. Furthermore, the unique ability of the in-situ analysis of cell interactions we can provide with our approach may yield a true representation of the network, producing unexpected results such as the mutual avoidance of receptor-ligand bearing T cells and macrophages in uterine leiomyosarcomas (Manzoni M, et al., 2020). Finally, a number of recent papers investigates the immune microenvironment focusing on response to therapy rather than prognosis and survival as main outcome. Sade-Feldman et al., 2018, identify two major CD8^+^ cell states, one with increased expression of genes linked to memory, activation and cell survival and one with enrichment in genes linked to cell exhaustion. A higher amount of memory/active cells at the baseline was found in responders to checkpoint inhibition, while exhausted CD8^+^ T cells were more abundant in non-responders. Also, in their data set, TIM3, together with CD39, marked stronger than other genes the state of exhaustion of the CD8^+^ T cells. Active cytotoxic T cells as a hallmark for response to checkpoint inhibitor therapy were also found by Riaz et al., 2017, whose work showed not only an increased number of TILs, NK and M1 macrophages in responders as a consequence of Nivolumab administration, but also enhancement of cytolytic pathways genes, possibly a bystander for cytotoxic T cell activation. Going beyond single-cell analyses, Prat et al., 2017 uses a bulk digital mRNA expression method to define 12 signatures associated with response and progression-free survival in different types of cancer, including melanoma. Among these signatures, T cell activation (both CD4 and CD8) and IFN activation are included. Ayers et al., 2017 used the same bulk digital mRNA expression platform to develop a predictive “interferon-gamma” gene signature, and observed that tumors with low signature scores characterized non-responders.

All of the points above have been added to the Discussion section.

2) Neighbourhood analysis. The reviewers agree that this is one of the most interesting results in this manuscript as it adds information that other non-spatial single-cell-based are not able to provide.a) Please place relevant cell-cell interactions (e.g. melanoma:T-cell) in the main text instead of the supplementary materials, and mention panels C and D in the Results section.

Supplementary Data Figure 3 containing all the relevant melanoma cell-other cells interactions has been integrated in Figure 4 in order to appear in the main text. Panels C and D have been added in subsection “Neighborhood analysis”.

b) Please improve Figure quality by increasing font size and adding legends, and clarify why some cell types (e.g., TIM3 + cDC2) are not shown in Figure 4C.

Legend has been added in Figure 4, the font has been enlarged, all cell types are now kept constant in all the 4 schemes (before only the ones involved in relevant interactions were shown.

c) Please discuss how the fact that the "active" and "brisk" plots are nearly identical fits with the paper's main conclusion that the "brisk" state consists of both active and exhausted.

It is true that the neighbourhood analysis profile of the brisk cases was very similar to that of active cases and that non-brisk cases showed more cell-cell interactions linked with immune suppression. Nevertheless, both categories were not totally overlapping with the active and exhausted plots, but rather presented also some of the immune-stimulating and immune-suppressive interactions present either in the active of exhausted plots (eg: CD69+TIM3+Th interaction with anergic Tcy, common between brisk and exhausted, or the active Th towards the active Tcy present both in the active and non-brisk group). This is in line with the fact that the morphological categories are functionally heterogeneous. Moreover, the fact that brisk and active plots are similar can partially explain the good prognosis of the brisk cases, together with the fact that in brisk cases the melanoma cells are more in contact specifically with active/transition Tcy than in non-brisk cases, that instead have more exhausted and anergic cells in contact with melanoma cells. These considerations have been rephrased in the Discussion section and explained better in the Results section.

3) It is unclear what the qPCR and proteomics experiments add to the results. Please clarify how the fact that 50% of the brisk and 43% of the non-brisk tumours can be classified as "active" fits with the neighbourhood analysis, particularly point 2C above.

Since our TMA is composed exclusively of primary melanomas, one may object that in a metastatic setting cases with mainly active TILs may not be detected, maybe because the immune system of the patient is failing in keeping control of the tumor. Or maybe, in the metastatic setting to see a diffuse TILs infiltrate in a metastatic nodule would have been possibly correlated with real activation and not with exhaustion, and only morphological evaluation could be enough to evaluate the activation status of the TILs. To avoid to discover at a later research stage that in the metastatic stage our findings do not apply, the qPCR experiment was intended as a quick, explorative way to verify that the same functional classification in active and exhausted not only existed also in the metastatic setting, but also did not correspond to the brisk and non-brisk morphological classification. The idea was to check that the principles that we describe may be valid for immunotherapy, that it is generally applied only if the patient develops metastasis (even if since very recently adjuvant immunotherapy is starting to be introduced in the clinics). The brisk/non-brisk definition applies only to primary melanomas, but here in the metastatic setting we expanded the definition to metastasis classifying them as brisk if diffusely infiltrated by TILs and non-brisk if only partially infiltrated by TILs. The percentages could be influenced by the small amount of metastasis included for the analysis and are therefore not informative – what is informative here is only the fact that we can find cases mainly active and exhausted also in metastases, also using other methods, and they do not cluster according to the brisk-non-brisk paradigm. This has been clarified in subsection “qPCR and shotgun proteomics”.

4) Please add more explanation in this manuscript about what the MILAN technique entails, given that it does not seem to be in widespread use.

The MILAN technique has been published in detail in 2017 in the JoHC (doi.org/10.1369/0022155417719419), the manuscript has been downloaded >8,000 times and is now used in four labs. It is a versatile, robust, cheap technique, able to stain a single routinely processed section with >100 antibodies over >30 staining and stripping cycles, without antibody prioritization or cell loss. More specifications about the techniques and an additional reference has been added in subsection “Multiplex-stripping immunofluorescence”.

5) Principal component analyses. Please clarify the following points:a) Are the PCAs in Figure 2 created with all markers or just the select markers shown in Figure 2?

Figure 2 has been created using only CD8 positive cells (indicative of T cells) and the 4 markers relevant for their functional status: CD69, OX40, LAG3, and TIM3 (added to Figure 2 legend).

b) For the classification of cells into the "active" and "exhausted" states, were all cells from all patients put together in the analysis, and if so, did PC1 divide cells by patient? What variability did the axis of maximum variation capture?

For the classification of cells into “active” and “exhausted” cells from all patients were put together in the analysis. PC1, did not show a differential expression of any of the 4 markers in the rotation matrix as all of them were negative (see the table in Figure 2—figure supplement 4). PC1 explained 39,39% of the total variance and was not associated to a patient effect as shown by the Figure below (no batch effect due to patient). If we visualize a 3D scatter plot with PC1, PC2 and PC3 with every cell colored by activation we can see that the resulting 3D structure has a pyramidal shape with PC1 parallel to its main axis and its apex (main vertex) found at the maximum value of PC1. As seen in 3D plot and supported by the projections on Figure 2C, points near the vertex (centroid of the projection in Figure 2C), do not show a strong expression for any marker. On the other hand, points close to the base of the pyramid (low values of PC1) show a strong differential expression of the activation and exhaustion markers. Therefore, PC1 was disregarded for the activation/exhaustion analysis and only PC2 and PC3 were used. Figure 2—figure supplement 3 and legend have been updated.

**Author response image 12. respfig12:** 

6) Correlation between core status and tumour regression. Please answer the following questions:a) How is a late regression area defined?

According to Botella-Estrada et al., 2014 “The presence of regression was evaluated according to criteria previously established by a detailed review of the dermatologic literature and our own experience as follows: (1) Small or large areas with a decrease or an absence of melanoma cells in the dermal component of the tumor (2) Fibrosis (3) Inflammatory infiltrate (4) Melanophages (5) Neovascularization (7) Epidermal flattening (8) Colloid bodies (apoptosis of keratinocytes/melanocytes). Early regression was defined by criteria 1 (usually small foci), 2 (to some degree), and 3 (dense inflammatory infiltrate) with any combination of criteria 4 through 7. Late regression was defined as criteria 1 (usually medium to large areas) and 2 (very evident) with any combination of criteria 3 through 7.” This was added to subsection “TMA construction”.

b) Do the authors think the reported correlation may have arisen by chance, given that No Regression vs Early regression (comparison of the most different states) did not show any differences? Were the differences that were detected in the expected direction (i.e. early regressed tumours had a higher activation status than late regressed tumours)?

In this particular case, the number of early (8) and late (5) regression patients is significantly smaller than the number of no regression patients (33). Thus, a larger cohort may be needed to validate these results. However, first of all, considering the sign of the reported differences (late regression more active than early regression and no regression) we do think that the reported results are coherent with the real biology. Early regression is a controversial entity among pathologists: in spite of the criteria mentioned above, there is no agreement about the fact that it represents a real regression phase or rather a very pronounced brisk infiltrate. In this view, the dense inflammatory infiltrate in early regression may actually lead to late regression or may not, getting exhausted and disappearing without having actually a tangible anti-tumoral result. In agreement with this controversy, our data shows not only that early regression it is a very heterogeneous group displaying a great variability in activation levels (Figure 3A), but also that overall it has lower activation levels then late exhaustion (Figure 3B). Moreover, this data seems to be supported by neighborhood analysis. In late regression, a network of activating interactions between active Tcy and active Th was prevalent over few immune suppressing interactions (specifically, the ax between Treg-CD69+TIM3+Th-Exhausted Tcy) (Figure 3C), while in early regression the dense infiltrate contains a more complex network of interactions with many of them contributing to immune impairment and, therefore, to the detected lower levels of activation. From Figure 3D it is possible to see that the interactions between active Th and active Tcy disappeared in this group, to leave space to aggregates of B cells located in strict proximity with anergic, proliferating and active T cells and probably stimulated by TIM3+cDC2, counterbalancing the effect of the immune stimulation between cDC1 and active Th. These data support the relationship between activation and late, but not early, regression. This discussion has been added to subsection “Neighborhood analysis” and to the Discussion section. Figure 3 has been added.

7) The reviewers agreed that it is necessary to add a section in the discussion regarding how the authors' results compare and fit with previous publications, in particular Sade-Feldman et al., 2018; Ayers et al., 2017; Prat et al., 2017; Riaz et al., 2017 and Tirosh et al., 2017.

The papers suggested here came across in the PubMed search while looking for more articles to complete point 1c, therefore we have discussed them above.

8) Regarding statistics in general. The reviewers suggest displaying p-values in all Figures where there have been statistical tests and justify the use of the chosen tests. (For example, reviewers mention that authors could have used logistic regression to analyze the relationship between their variables, as well as ANOVA instead of t-tests with Holm's or FDR corrections). Clarifications about when multi-testing correction was applied (or not) should be added.

P-values have been added in all the Figures where there have been statistical tests. The following table includes the justification for the test chosen in each case (this table has been added to the manuscript as Supplementary file 5 with the clarification if multi-testing correction was applied, Materials and methods section):

[Editors' note: further revisions were suggested prior to acceptance, as described below.]The reviewers agree that the manuscript has been substantially improved but there are some minor remaining issues that need to be addressed before acceptance, as outlined below:Could you please add some discussion on the following:1) To assign an activation state to patients, the authors didn't combine states per core but per cell – Doesn't this depend on the number of cells that were successfully assessed? Reviewers would like to see an acknowledgment that perhaps sampling more/less cells per patient could have an influence in this classification.

We acknowledge that in this analysis, like in every statistical test, the exact p-value varies together with the sample size. In order to evaluate the robustness of our patient classifications into “Active”, “Exhausted”, and “Transition” we reclassified each patient using different sampling sizes (from 10 to 1000 cells using steps of 10) 100 times. This resampling analysis is summarized by Author response image 13, below, and it shows that our classification of patients into “Active”/”Exhausted” is very robust, requiring a relatively small number of cells to classify each patient reliably. We evaluated for every patient the minimum number of cells required to obtain the same significant classification in at least 95% of the simulations. The average value across all patients is 180 cells for active cases, and 174 for exhausted cases. Some cases, (patients 2, 3, 8, 12, 15, 18, and 21) required as little as 30 cells to obtain the same significant classification in at least 95% of the simulations. Only 3 cases (7, 9, and 17) required a relatively large number of cells (>360, ~2 times the average) to obtain a significant classification in at least 95% of the simulations. Author response image 13 also suggests that some patients classified as “Transition” could have been identified as “Active”/”Exhausted” would the number of identified Tcells had been bigger. We have added this paragraph to the Materials and methods section.

**Author response image 13. respfig13:** Resampling analysis. CD8+ TCell populations were sampled 100 times for different sampling sizes (10 to 1000 with a step of 10). For every simulation, we reclassified the patient into “Active”/”Transition”/”Exhausted” using the approach described in the methods. This analysis shows the robustness of the patient classification methodology with most of the patients requiring a very small sample size in order to obtain at least 95% of consistent classifications.

2) Can the authors please expand a little on the logic behind classifying "non-brisk with exhaustion" as poor prognosis one whereas "brisk with exhaustion" is classified as good prognosis? How do the authors define the relationship between these two different classification methods? Is morphology then also important to take into account even when the single cell status is being considered? (And, was this the comparison that got the better p-value? Were the other possibilities (e.g. classifying brisk with exhaustion as poor prognosis) also considered?

The reviewer refers to the explanation given in the response letter. Yes, other possibilities were considered but brisk patients with activation, brisk patient with exhaustion, and non-brisk patients with activation were grouped together because there were no deaths from these groups in our cohort, therefore the Kaplan Meier curves of these 3 groups were overlapping, as shown here:

**Author response image 14. respfig14:** 

The authors consider the two classifications to be complementary, since the main finding of this analysis suggests that combining the two classification parameters achieves a better definition of the prognosis.

The reviewers would also like to see clarification for one of the points in the rebuttal letter, please. One of them wrote, "The two explanations for categorising the "transition" patients into active or exhausted (p values 0.053 or 0.079) sound exactly the same to me. What is different? (However, I do not think this is mentioned in the main text)."

The first categorisation of “transition” patients in “active” or “exhausted” (corresponding to p-value = 0.053), reclassifies only those patients identified as “transition” with the original methodology (comparison with the background distribution). The second categorisation (corresponding to p-value = 0.079) does not use the original methodology and classifies all the patients based on the average level of activation of all its cells.

As the reviewers pointed out, this extra explanation was included in the rebuttal letter for further clarification, but the authors did not consider it relevant enough to include it in the main text unless the reviewers would like to see it specified in the main text. Would the reviewer prefer to see it in the main text?